# An open-source closed-loop Virtual Reality system to investigate social interactions and collective behavior in fish

Stéphane Sanchez[1], Ramón Escobedo[1,2,3,4], Renaud Bastien[1,2], Boris Lenseigne[5], Audrey Denis[2], Mathieu Moreau[2], Maud Combe[2], Andrew D. Straw[6,7], Clément Sire[4], Guy Theraulaz[2]*

1 Institut de Recherche en Informatique de Toulouse (IRIT), Université Toulouse Capitole, Toulouse, France, 2 Centre de Recherches sur la Cognition Animale, Centre de Biologie Intégrative, CNRS, Université de Toulouse III – Paul Sabatier Toulouse, France, 3 Laboratoire de Physique Théorique, CNRS, Université de Toulouse III – Paul Sabatier Toulouse, France, 4 Departamento de Matemáticas, Universidad Carlos III de Madrid Leganés, Madrid, Spain, 5 Izital BV, Delft, The Netherlands, 6 Institute of Biology I, Faculty of Biology, Albert-Ludwigs-Universität Freiburg, Freiburg, Germany, 7 Bernstein Center Freiburg, Albert-Ludwigs-Universität Freiburg, Freiburg, Germany

* guy.theraulaz@utoulouse.fr

## Abstract

This study introduces a low-cost, open-source, immersive Virtual Reality (VR) system designed to investigate real-time social interactions and collective behaviors in fish. Understanding collective animal behaviors, such as schooling in fish, presents significant observational challenges due to rapid and complex interactions. To address these difficulties, we developed an innovative closed-loop VR environment allowing precise control and measurement of interactions between real and virtual fish. This setup incorporates high-speed 3D tracking, real-time visual feedback, and automated data processing, creating realistic and interactive scenarios. We present experimental results, obtained with rummy-nose tetras (*Hemigrammus rhodostomus*), showing that freely moving real fish consistently adjusted their speed, depth, and spatial positioning to follow virtual fish effectively. Fish matched moderate virtual speeds comfortably, struggled slightly at higher speeds, and actively maintained vertical positioning to sustain group cohesion. The implementation of this VR system provides critical insights into sensory and cognitive processes underlying collective movement and decision-making in fish. By revealing both flexibility and constraints in fish behavior, our findings underscore the potential of VR technology to study social interactions in controlled yet realistic settings. Ultimately, this VR system advances our understanding of collective animal behavior, providing a foundation for cross-species studies linking neural processes, behavioral interactions, and group dynamics.

**Data availability statement:** Data from all the experiments are available from the figshare repository (https://doi.org/10.6084/m9.figshare.29097605). The codes for the 3D tracking, the trajectory, and the rendering of virtual fish are available from the figshare repositories (https://doi.org/10.6084/m9.figshare.30188836.v1 (Acquisition and 3D tracking software)) (https://doi.org/10.6084/m9.figshare.30188842.v1 (Trajectory simulator)) (https://doi.org/10.6084/m9.figshare.30188845.v1 (3D anamorphosis rendering software)) (https://doi.org/10.6084/m9.figshare.30188839.v1 (Calibration routines)).

**Funding:** Agence Nationale de la Recherche (ANR-20-CE45-0006-1) Spanish State Research Agency, AEI-Agencia Estatal de Investigación grant PID2020-115088RB-I00.

**Competing interests:** The authors have declared that no competing interests exist.

## 1 Introduction

Understanding collective animal behavior has long been a fundamental goal in behavioral biology and neuroscience [1,2]. Fish schools, bird flocks, and insect swarms exhibit remarkable coordination that emerges from local interactions among individuals [3,4]. These interactions are crucial in shaping movement dynamics, predator avoidance, and decision-making processes in groups [5,6]. However, studying these interactions in natural conditions presents methodological challenges due to the complexity of tracking individual behaviors and measuring their real-time influence on the group. These constraints explain the need to carry out studies in controlled environments to decipher these interactions and cognitive processes involved at the individual scale [7–10]. Traditional observational methods and open-loop experimental setups often fail to capture the bidirectional influence between individuals and their environment. To address these limitations, virtual reality (VR) systems have emerged as powerful tools to investigate animal perception and behavior, allowing researchers to manipulate environmental variables and control social stimuli with high precision [11].

Recent advances in VR technology have enabled the study of visual-motor feedback in fish and other animals under open-loop and closed-loop conditions. For instance, VR studies on fruit flies have provided insights into how insects process visual motion and adjust their flight paths in response to virtual stimuli [12]. VR systems have also been employed to examine goal-driven navigation in zebrafish larvae by linking their tail movements to virtual environments [13]. Similarly, high-resolution VR platforms have been used to analyze neuronal activity in head-restrained adult zebrafish responding to dynamic visual stimuli [14]. Researchers have explored how animals respond to virtual predators, social cues, and environmental changes. For example, using a VR system for studying wild-caught coral reef fish, it has been demonstrated that these fish could distinguish between virtual predators, conspecifics, and non-threatening species [15]. In another study, [16] found that animals in VR could make complex navigational decisions based on visual cues, shedding light on how sensory information is integrated in the brain. Additionally, VR has been instrumental in studying threat avoidance in humans through the VRThreat toolkit [17]. This software simulates realistic danger scenarios, allowing researchers to examine human defensive behaviors in high-stakes situations. These studies have demonstrated that VR can be effectively integrated with neurophysiological techniques, allowing precise manipulation of sensory inputs and real-time tracking of behavioral responses [18].

Building on these foundations, recent VR-based research has provided new insights into how fish process social cues and make decisions in collective contexts. It has been shown that zebrafish do not simply average the directions of neighbors but can explicitly choose which conspecific to follow, producing bifurcations in their trajectories when confronted with multiple leaders [19]. Other studies demonstrated that larval zebrafish transform simple retinal occupancy statistics into motor responses, with a developmental transition from repulsion to attraction as cohesive

shoals form [20]. Immersive VR has been used to reveal that juvenile zebrafish follow conspecifics according to a proportional-derivative pursuit law, robustly capturing pursuit dynamics across single- and multi-target scenarios [21]. Earlier work already highlighted the potential of immersive VR to study startle responses [22], and further showed how larvae chain discrete swim bouts into prey-capture sequences under closed-loop visual feedback [23]. Technical developments such as panoramic projection systems have expanded the capacity for wide-field, high-resolution stimulation [24]. More recently, closed-loop VR combined with two-photon calcium imaging revealed that some zebrafish generate neural ensembles encoding prediction errors, enabling them to escape more efficiently—evidence that vertebrates integrate both reward maximization and surprise minimization strategies when making escape decisions [25]. Together, these studies illustrate how VR provides a unique window into the sensory-motor and cognitive mechanisms that govern collective behavior, from elementary visuomotor transformations to higher-order decision-making.

In this article, we introduce an innovative and low-cost closed-loop VR system specifically designed to study real-time social interactions among fish. Our objective is to demonstrate how this technology allows precise control and measurement of behaviors, providing new insights into the mechanisms driving collective movement. This experimental platform integrates high-speed tracking, real-time visual feedback, and automated data processing to measure how freely swimming fish dynamically respond to computer-controlled virtual conspecifics. By simulating realistic social interactions within a VR environment, our system allows us to quantify the influence of local neighbor interactions on collective movement and decision-making. Unlike traditional open-loop VR systems that impose predetermined stimuli, this closed-loop design continuously updates the visual stimuli based on real-time behavioral feedback, ensuring a dynamic and interactive experimental paradigm [12].

One of the key advantages of this system is its ability to simulate controlled yet biologically relevant interactions that influence group coordination. Previous research has shown that synchronized movements involve selective social interactions between fish and that vision plays a crucial role in behavioral decisions [26–29]. By systematically varying the behavioral parameters of virtual conspecifics, we can investigate how sensory feedback modulates movement decisions, spatial alignment, behavioral contagion, and leadership emergence within groups.

We detail the technical components, validate our closed-loop VR system through controlled experiments, and present preliminary findings obtained with Rummy-nose tetras (*Hemigrammus rhodostomus*). This species has a strong tendency to school, with an intermittent swimming mode characterized by alternating bursts and coasting. While our experiments focus on interactions involving one real fish and one virtual fish, the system can easily include several virtual fish and adapt to other species by modifying the 3D shape and color of the virtual fish and their behavioral and swimming patterns. Ultimately, our VR system aims to clarify how freely moving fish perceive and integrate local social cues, enhancing our understanding of collective behaviors at both neural and behavioral levels.

## 2 Materials and methods

### 2.1 Experimental setup

The experimental setup is similar to the one used in FishVR [11]. The main differences are a larger bowl to accommodate adult rummy-nose tetras (3 to 4 cm in length) and a single depth camera as a capture device (Fig 1). The acrylic bowl is made by Immersive Display Group, London, UK. It is a subsection of a sphere filled with 15 L of water, resulting in a depth of 14.6 cm and a diameter of 52.2 cm at the water surface. Infrared light for tracking is provided by 8 IR lamps (wavelength of 850 nm) from underneath the bowl. The tracking camera is an Intel RealSense D435 depth camera [30]. It is positioned above the water surface, at 47.9 cm above the center of the water surface. IR filters are applied to the two IR sub-cameras of the camera. A single computer (Dell Precision 3640 Tower Workstation PC with a NVIDIA RTX 3070 graphics card) does all the real-time computations: tracking, trajectory simulations, virtual environment computation, and display. A DLP LED video projector (OPTOMA WUXGA) illuminates and displays the virtual environment on the bowl from beneath.

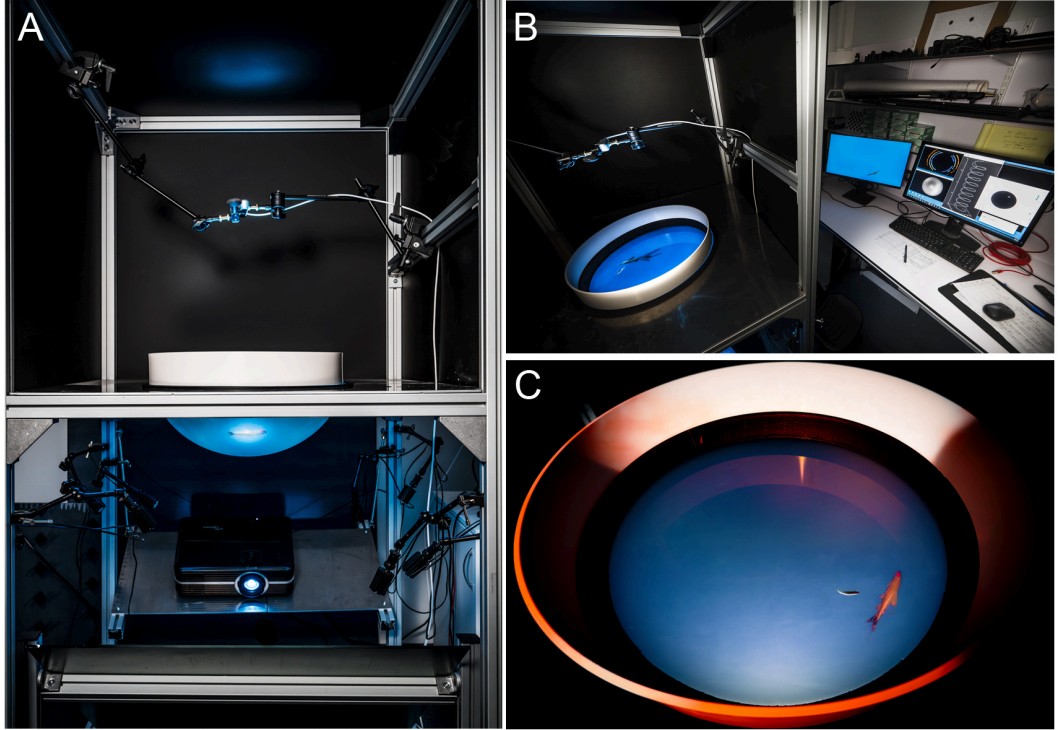

**Fig 1**. **Analysis and modeling of interactions between schooling fish in a closed-loop virtual reality setup.** (A) Closed-loop virtual reality setup used to measure and analyze in real time the interactions between fish within a school and their effects on individual behavior. (B) The virtual reality setup makes it possible to track in real time the 3D movements of a fish moving freely in a hemispheric tank. (C) These data then feed a mathematical model controlling the behavior and movement of one or several realistic virtual fish of the same species, the 3D anamorphic image of which being projected onto the hemispherical screen constituting the tank wall.

Closed-loop interactions between real fish and virtual ones require 3 separate software elements:

1. Acquisition and 3D tracking software that gets frames from the Intel RealSense D435 camera and tracks fish positions from them.
2. Trajectory simulator that simulates virtual fish behaviors, with or without interactions with real fish.
3. Rendering software that displays virtual fish according to the simulated positions and the real fish position (to perform anamorphosis rendering).

Network UDP requests transfer positions of fish (real and virtual ones) from one software to another.

## 2.2 Images acquisition and real-time 3D tracking of fish

### 2.2.1 Images acquisition.
In our experimental setup, we use an Intel RealSense D435 camera to acquire images of the real fish freely moving in the bowl. This 3D camera is autocalibrated and features two IR cameras, one RGB camera, and a laser projector. For each camera, it provides a video stream and a computed depth map. The acquisition software is based on RealSense SDK [31] to control the D435 camera and to acquire, at a rate of 30 to 90 frames per second, both IR frames (left and right) and one depth map aligned with the left IR frame.

**2.2.2 3D tracking.** The 3D tracking software is written in Python and can operate in two different modes, depending on which camera provides the information it uses (Fig 2):

1. Using directly the depth map. This offers the advantage of directly obtaining the real-world $z$ coordinate of any tracked pixel $(i,j)$ in the depth map image (the RealSense SDK automatically converts any pixel $(i, j)$ into real-world $(x, y, z)$ coordinates).
2. Using the standard stereo vision from both the IR left and right frames. Each frame is processed with OpenCV [32] routines (mainly masking and background subtraction routines) to isolate real fish pixels.

The advantage of using a depth camera is that the frame processing required to compute an accurate depth map is performed directly on its hardware, and is therefore theoretically much faster than external computation. Our first data sets showed surprising regular bands in the depth distribution of real fish positions, as if the fish only swam in discrete regular depths. Further investigations have shown that some disparities (disparity refers to the distance (in pixels) between two corresponding points in the left and right frames of a stereo pair) from D435 seem to be rounded to the nearest integer value, as shown in Fig 3A. Implementing our own stereovision computation from both IR frames solved the depth distribution problem (cf. Fig 3A). But, as shown in Fig 3B, raw depth data is much noisier with stereo vision than data from the depth camera. To mitigate this noise, computed depth using stereovision is smoothed with a Kalman filter (as implemented in Python).

Tracking of individual fish is based on the raw data provided by the RealSense camera. Input is therefore a 16-bit depth map with a resolution $848 \times 480$ for immediate 3D tracking or two 10-bit $848 \times 480$ IR frames for stereo vision. Masking

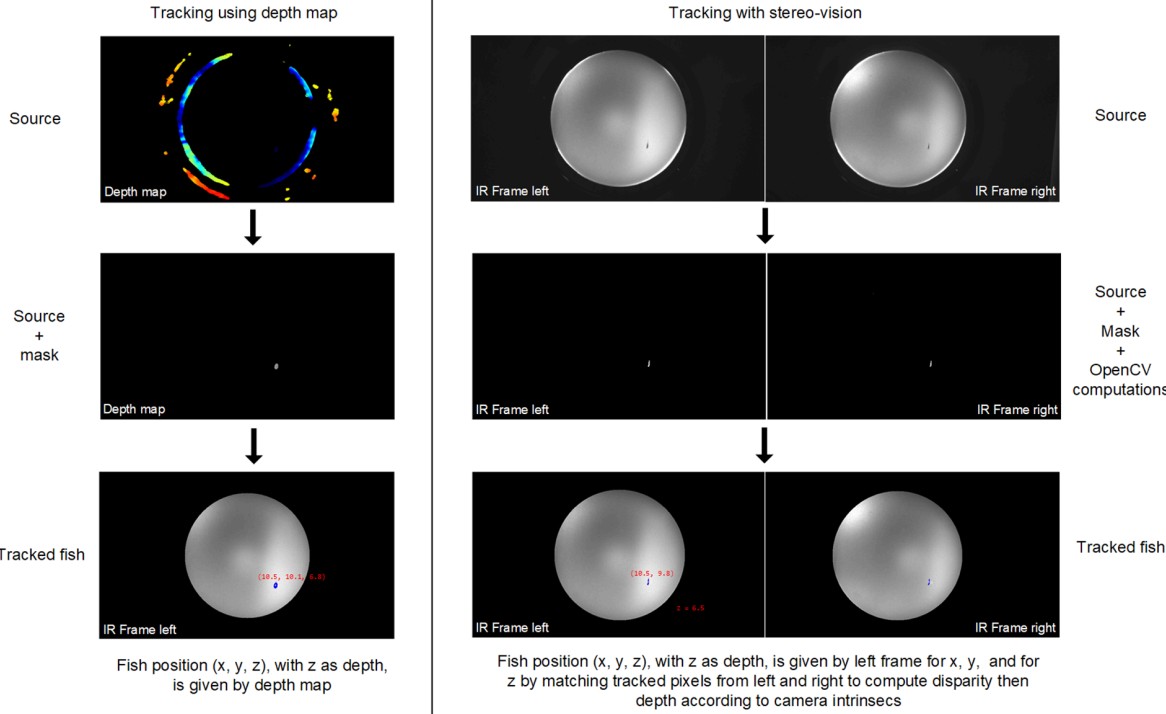

**Fig 2. Operating modes of the 3D tracking.** Detail of the tracking process for the same frame set using D435 camera computed depth map (left) and classical stereo-vision computation from left and right infrared frames (right).

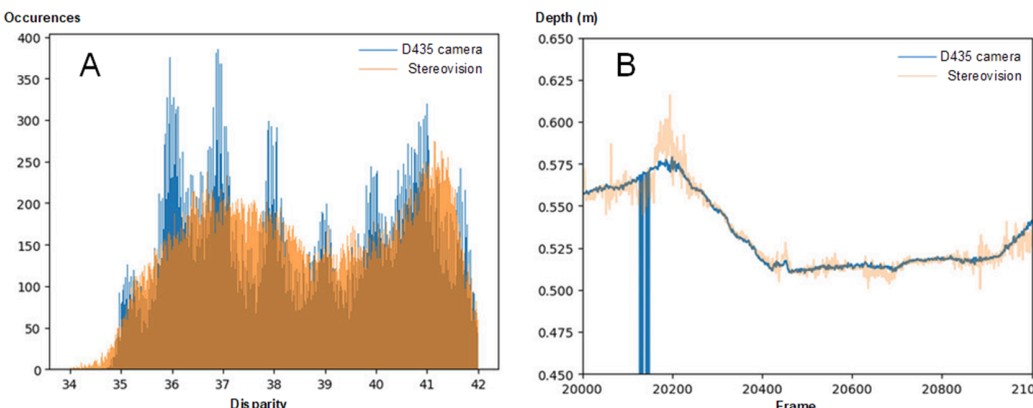

**Fig 3**. **Disparities between the depth map and the stereo vision tracking modes.** (A) Disparities computed with D435 camera (blue) and standard stereovision (orange) for 10 min of tracking data. (B) Raw depth for a 1000-frame sample in the same experiment.

is performed on each data frame so that only targets within the bowl and within the desired depth ranges are considered. From the depth map, filtered data is delivered to the tracker as 3D points or, from the IR frames, as 2D points. Tracking is performed using a Random Matrix Tracker [33], [34]. This specific kind of filter is a derivation of the well-known Kalman filter, most suited to the tracking of targets represented as a point cloud. It is akin to a traditional Kalman filter with the addition of an extension matrix that represents the spread of the point cloud as the tracking goes on. Each target/fish has its own tracker attached.

The target is represented by a state vector that contains the target's dynamics up to a specific order (order two in this presentation: position and speed in 2D or 3D): $x_k^\mathsf{T} = [r_k^\mathsf{T}, \dot{r}_k^\mathsf{T}]^\mathsf{T}$, at time step $k$, where dot denotes derivation with respect to time. The spread is a Symmetric Positive Definite matrix: the covariance matrix $\mathbf{X}_k$ of the target. The measurement vectors $y_k^j$ from multiple measurements reflect the position and spread of the target at the time step $k$. The tracking is performed in the reference frame of the camera image; therefore, the measurement matrix is : $\mathbf{H} = [\mathbf{I}_{d\times d}\mathbf{0}_{d\times d}]$. The measurement of the point cloud at time $k$ is described as $n_k$ points with a covariance $\mathbf{X}_k$: $\mathbf{Y}_k = \{y_k^j\}_{j=1}^{n_k}$. Since the point cloud extension $\mathbf{X}_k$ covers the spread of the measurements, we neglect more specific sensor noise. With these settings, the prediction equations become:

$$x_{k|k-1} = \mathbf{F}x_{k-1|k-1},$$
$$\mathbf{P}_{k|k-1} = \mathbf{F}\mathbf{P}_{k-1|k-1}\mathbf{F}^T + \mathbf{Q},$$
$$\mathbf{X}_{k|k-1} = \mathbf{X}_{k-1|k-1},$$
$$\alpha_{k|k-1} = 2 + \exp\left(\frac{-T}{\tau}\right)(\alpha_{k-1|k-1} - 2),$$

where $\tau$ is a constant related to the target's agility (higher $\tau$ means less agile target), $T$ is the time prediction interval, $\mathbf{F}$ is the process model that assumes constant speed, and $\mathbf{P}$, $\mathbf{Q}$, and $\mathbf{R}$ are the state estimation, process noise, and measurement noise covariance matrices, respectively, as in a regular Kalman filter. The estimator next undergoes an update step based on measurements.

We start with the estimate of the measurement error variance considering the target spread and the sensor noise ($\lambda$ is an additional scaling factor),

$$\mathbf{Y}_{k|k-1} = \lambda \mathbf{X}_{k|k-1} + \mathbf{R},$$

then the innovation covariance and Filter gain,

$$\begin{aligned}
\mathbf{S}_{k|k-1} &= \mathbf{H}\mathbf{P}_{k|k-1}\mathbf{H}^\mathsf{T} + \frac{\mathbf{Y}_{k|k-1}}{n_k}, \\
\mathbf{K}_{k|k-1} &= \mathbf{P}_{k|k-1}\mathbf{H}^\mathsf{T}\mathbf{S}_{k|k-1}^{-1},
\end{aligned}$$

and then the remaining steps, which closely follow the conventional Kalman update for the state vector and estimation covariance:

$$\begin{aligned}
x_{k|k} &= x_{k|k-1} + \mathbf{K}_{k|k-1}(\overline{Y}_k - \mathbf{H}x_{k|k-1}), \\
\mathbf{P}_{k|k} &= \mathbf{P}_{k|k-1} - \mathbf{K}_{k|k-1}\mathbf{S}_{k|k-1}\mathbf{K}_{k|k-1}^\mathsf{T},
\end{aligned}$$

where $\overline{Y}_k$ is the barycenter of the measurement point cloud.

Next, the extension update includes the covariances induced by target dynamics and by measurement noise. The latter is computed explicitly as the covariance matrix of the measurement error with respect to the estimated state,

$$\mathbf{N}_{k|k-1} = (\overline{Y}_k - \mathbf{H}x_{k|k-1})(\overline{Y}_k - \mathbf{H}x_{k|k-1})^\mathsf{T},$$

and similarly for the measurement covariance,

$$\overline{\mathbf{Y}}_k = (\overline{Y}_k - \mathbf{Y}_k)(\overline{Y}_k - \mathbf{Y}_k)^\mathsf{T}.$$

We now have the necessary building blocks to compute the extension contribution of target's dynamics and measurement noise contribution:

$$\begin{aligned}
\widehat{\mathbf{N}}_{k|k-1} &= \mathbf{X}_{k|k-1}^{\frac{1}{2}}\mathbf{S}_{k|k-1}^{-\frac{1}{2}}\mathbf{N}_{k|k-1}(\mathbf{S}_{k|k-1}^{-\frac{1}{2}})^\mathsf{T}(\mathbf{X}_{k|k-1}^{\frac{1}{2}})^\mathsf{T}, \\
\widehat{\mathbf{Y}}_{k|k-1} &= \mathbf{X}_{k|k-1}^{\frac{1}{2}}\mathbf{Y}_{k|k-1}^{-\frac{1}{2}}\overline{\mathbf{Y}}_k(\mathbf{Y}_{k|k-1}^{-\frac{1}{2}})^\mathsf{T}(\mathbf{X}_{k|k-1}^{\frac{1}{2}})^\mathsf{T}.
\end{aligned}$$

Finally, once we have set $\alpha_{k|k} = \alpha_{k|k-1} + n_k$, the final state extension is written as a weighted sum of the different contributions:

$$\mathbf{X}_{k|k} = \frac{1}{\alpha_{k|k}}\left(\alpha_{k|k-1}\mathbf{X}_{k|k-1} + \widehat{\mathbf{N}}_{k|k-1} + \widehat{\mathbf{Y}}_{k|k-1}\right)$$

The tracker is initialized with the knowledge of the number $\nu$ of fish to track. We assume that the first dataframe contains each fish visible, preferably separated from each other. The initialization procedure aims at providing a realistic initial estimate of the position and spread of each fish.

Therefore, the following procedure is repeated for each $i = 1, \ldots, \nu$ fish:

1. Select a point from the data frame and initialize the tracker.
2. Select a few close points below on a Cartesian distance Threshold.
3. Repeat until convergence of the state estimate:
    i. Perform a measurement update run of the tracker based on the selected points.
    ii. Define a search area with center $x_{k|k-1}^i$ and covariance $\sigma = \mathbf{X}_{k|k-1}^i + \mathbf{P}_{k|k-1}^i + \mathbf{R}^i$.
    iii. Add points to the fish based on the Mahalanobis distance:

$$\delta_{\text{Maha}}^i(y_k^j) = \sqrt{(y_k^j - x_k^i)^T(\mathbf{X}_{k|k-1}^i)^{-1}(y_k^j - x_k^i)},$$

    with a threshold based on desired Chi-squared likelihood.
4. Remove the selected points from the data frame.

Once these steps are performed, the normal tracking loop takes place. Firstly, a prediction step of the tracker is performed for each target. Next, a new data frame is brought in. Data points are assigned to each target considering the predicted new state $x_{k|k-1}^i$ and extension $\mathbf{X}_{k|k-1}^i$ of each $i = 1, \ldots, \nu$ fish. We use a nearest neighbor strategy based on the Mahalanobis distance between the point and each candidate target/fish. Furthermore, a threshold is applied to a maximal acceptable distance based on a reasonable size for a fish. Once measurements have been attributed to each target, the update steps as described above can take place, leading to a corrected state $x_{k|k}^i$ and extension $\mathbf{X}_{k|k}^i$ of each fish. Fish clusters, occlusions, and other artifacts might, however, lead to scenarios where, for some time steps, no measurements are found for one or several targets. Tracking continues for these targets using the prediction equations only. If the situation persists for too long, a full reset of the filter is performed following the procedure used for initialization. Previously existing targets are attached to the closest new target in terms of Cartesian distance, and normal tracking resumes.

The tracker operates in real time at a frame rate of 30 to 90 frames per second. In our experiments, both process and noise covariances are set as identically scaled identity matrices. For each target, filter settings are as follows: reactivity is set to $\tau = 10$, the innovation scale factor is $\lambda = 1$, and $\alpha_{0|0} = 1$.

## 2.3 Trajectories simulator software

A C++ Qt [35] application simulates the trajectories of virtual fish. The application collects real fish 3D position and orientation from UDP requests sent by the 3D tracking software (at 30 to 90 requests per second) and, for each request, computes in real-time new positions and orientations of each virtual fish according to its simulated behavior. All orientations and positions are immediately sent via the UDP request to the rendering application. In the context of this paper, two behaviors are considered: passive (open-loop) circular trajectories and rhodonea trajectories (see Fig 4). However, our system also allows for the position and heading of the virtual fish to be controlled in closed-loop by a behavioral model (like the data-driven model of [10] for rummy-nose tetra) exploiting the current position and speed of the real fish. In particular, the parameters of such a behavioral model (comfort speed and depth, strength of the attraction and alignment interactions with the real fish...) can be modified in real-time in the Graphical User Interface (GUI) of our VR system (see Fig 4).

**Circular trajectories.** Circular trajectories represent a simple behavior in which virtual fish swim at constant speed and constant depth in circles centered within the bowl. The application allows for dynamic changes to the speed, depth, and radius of the swimming circle. Virtual fish position $(x_n, y_n, z_n)$ is given by

$$x_n = R\cos(\omega t^n), \quad y_n = R\sin(\omega t^n), \quad z_n = z_0,$$

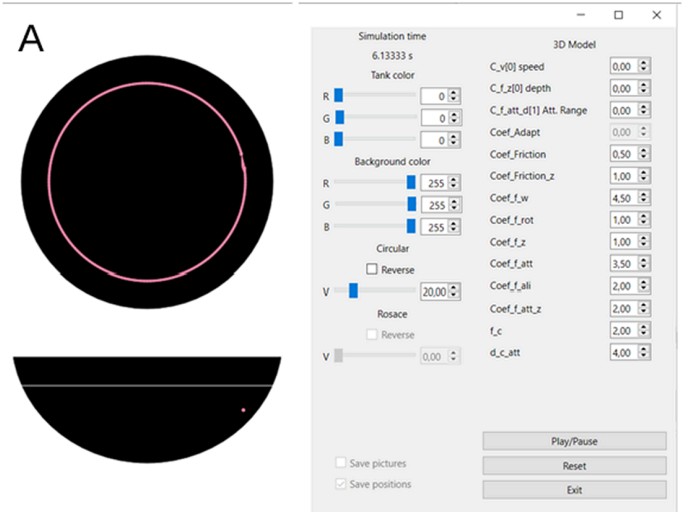
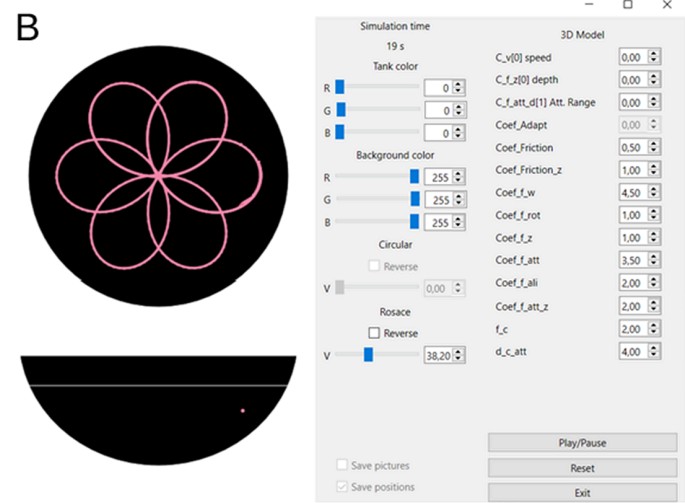

**Fig 4**. **Graphical User Interface (GUI) of the VR system.** Trajectory simulator performing (A) circle and (B) rhodonea trajectories. The GUI allows to visualize in real time the trajectory of the real and VR fish (in the $xy$ and $xz$ planes), and to modify instantaneously the parameters of the model driving the VR fish.

where $t^n = n\Delta t$ is the discrete time, $\omega = v/R$ is the angular speed, $R$ is the radius of the circle, $v$ is the constant linear speed along the circle, and $z_0$ is the fixed swimming depth. See Fig 4A.

**Rhodonea trajectories.** A rhodonea (also known as a mathematical rose) is a flat curve that has the shape of a flower with several petals (Fig 4B). We consider virtual fish performing a rhodonea curve centered within the bowl at a constant angular speed $\omega = v/R$ and constant depth $z_0$. As before, the application allows adjustments of the maximum radius $R$ of the petals and of the number of petals, determined by $n_{\mathrm{rod}}$: the number of petals is $n_{\mathrm{rod}}$ if $n_{\mathrm{rod}}$ is odd, and $2n_{\mathrm{rod}}$ if $n_{\mathrm{rod}}$ is even. The discrete position of the virtual fish at time step $t^n$ is given by

$$x_n = R\cos(n_{\mathrm{rod}}\,wt^n)\cos(wt^n), \quad y_n = R\cos(n_{\mathrm{rod}}\,wt^n)\sin(wt^n), \quad z_n = z_0.$$

### 2.4 VR-fish rendering

A C# Unity [36] application renders the virtual environment and virtual fish. From UDP requests sent by the trajectory simulator (at 30 to 90 requests per second), the rendering application gathers positions and orientations of both real fish and virtual ones. The real fish's position and orientation allow for the computation of anamorphic perspective, while the virtual fish's positions and orientations enable them to be rendered in accurate places in the bowl relative to the real fish. The rendering of virtual fish uses a realistic animated model of the Rummy-nose tetra (Fig 5). It is nevertheless possible to assess different fish species by substituting the 3D model with an appropriate one or, in the context of investigating non-social stimuli, by employing geometric shapes (e.g., spheres or cubes) in place of fish models.

**2.4.1 Calibration and projection mapping.** The basis of the Fish VR system is to project a rendering of a virtual environment on a surface, taking into account the perspective of individuals (anamorphosis) dynamically, using real-time tracking [11]. To render this projection of the fish's point of view accurately, the system's calibration is critical. This means that the position of the projection screen needs to be known exactly relative to the position of the fish. In the following, the 3D tracker's coordinate system serves as the reference space for all measurements.

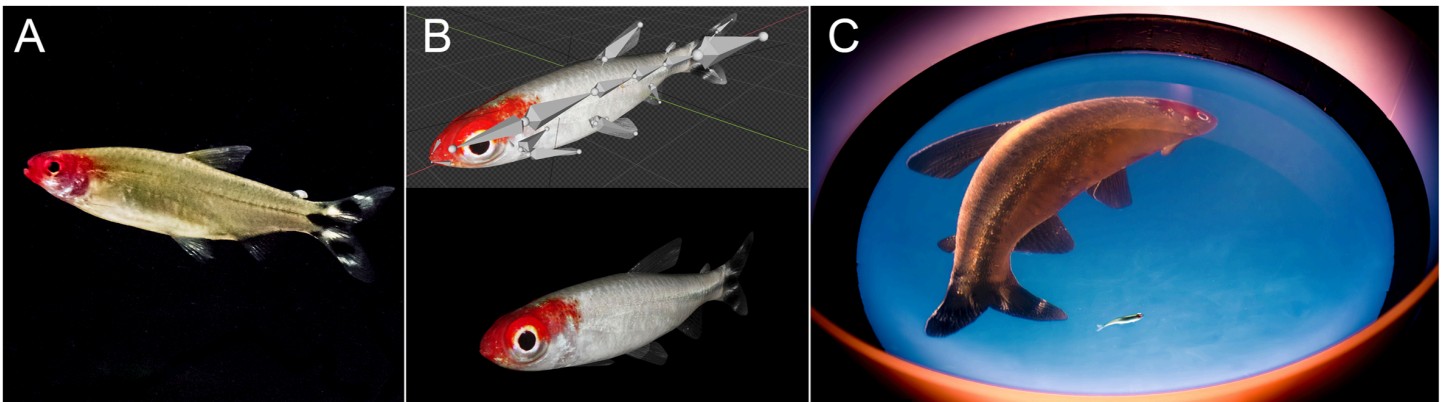

**Fig 5**. **Virtual fish 3D rendering.** (A) Real Rummy-nose tetra, (B) 3D animated model, and (C) anamorphic rendering of the virtual fish projected onto the acrylic bowl by the rendering application according to the 3D position of the real fish.

The process to calibrate the Fish VR requires one to get the following elements:

1. the position and geometry of the projection screen, i.e., the bowl, in the reference space,
2. the mapping between the projector's pixels and their corresponding 3D locations on the bowl's surface, and
3. the position of the water surface in the reference space.

Using these calibration data, combined with real-time tracking of the fish, the system can dynamically calculate and render the appropriate virtual scene onto the bowl, pixel by pixel, simulating the view from the fish's perspective.

To simplify calibration, the *z*-axis of the camera is aligned beforehand with the direction of gravity. This is not a necessary step, as the direction of the camera in real space can be recovered from the direction of the water surface. This simplification is allowed by the use of a single camera, compared to previous Fish VR setups [11], and will be assumed for the rest of the calibration.

**3D scan of the bowl.** The first step of calibration is to recover the position and geometry of the 3D bowl in space. The RealSense D435 is a stereoscopic depth camera that can offer a 3D mesh of the photographed scene. The accuracy of the stereoscopic camera can be increased when used with the embedded projecting laser. The scanning process consists of 5 steps:

1. Ten 3D images of the bowl are acquired.
2. The images are thresholded by distance to remove objects such as the table holding the bowl.
3. The images are then averaged to reduce the noise.
4. This 3D map is then directly fitted with a sphere.
5. A 3D image of this fitted sphere is synthesized 3D image from the point of view of the camera.

The geometry and position of the bowl are then acquired, namely, the position of the center of the sphere relative to the camera, $\mathbf{x}_{bowl} = (x_{bowl}, y_{bowl}, z_{bowl})$ and the radius of the sphere $R_{bowl} \sim 25\,cm$.

**Projection mapping on the sphere.** The second step is to establish a mapping that links each pixel coordinate in the projector's output image to its corresponding 3D point on the surface of the calibrated sphere (the bowl). Each given projector pixel coordinate, $\mathbf{l}_P = (i_P, j_P)$, should be associated with the spatial position of its projection on the bowl, $\mathbf{x}_P = (x_P, y_P, z_P)$.

**Mathematical description of the mapping.** To simplify the calculations in this section, we consider the referential where the bowl is at position $\mathbf{x}'_{\text{bowl}} = (0,0,0)$, and the radius of the bowl is $R'_{\text{bowl}} = 1$ such as:

$$\mathbf{x}' = (x', y', z') = \frac{1}{R_{\text{bowl}}}(\mathbf{x} - \mathbf{x}_{\text{bowl}}). \tag{1}$$

The direction of a light ray coming out of the pixel $(i_P, j_P)$ of the projected image is given by

$$\mathbf{w}_{ij} = (-\sin\phi'_i \cos\theta'_j, \; \sin\theta'_j, \; \cos\phi'_i \cos\theta'_j), \tag{2}$$

where $\phi'_i = \phi_0 + i\Delta\phi$, $\theta'_j = \theta_0 + j\Delta\phi$, and $\phi_0$ and $\theta_0$ is the direction of the projected image center. Here, $\Delta\theta$ and $\Delta\phi$ are the differences of angle given by 2 consecutive pixels in the horizontal and vertical directions. Pixels are assumed to be square, so $\Delta\theta = \Delta\phi$.

Image dimensions are usually even, so that the center of the image is not exactly on one pixel. Using a Full-HD projection ($1920 \times 1080$), we consider the range of horizontal pixels $i_P$ to be $[-959.5, 959.5]$, while the range of vertical pixels $j_P$ is $[-539.5, 539.5]$.

The 3D positions $\mathbf{x}_{ij}$ in space of each pixel $(i_P, j_P)$ on the projection screen are defined as:

$$\mathbf{x}'_{ij} = (x'_{ij}, \; y'_{ij}, \; z'_{ij}). \tag{3}$$

As these points are distributed on the bowl, we have $(\mathbf{x}'_{ij})^2 = 1$ for all $ij$. The points are related to the position of the projector by the direction of the associated light ray $\mathbf{w}_{ij}$,

$$\mathbf{x}'_{ij} = \mathbf{x}'_P + d_{ij}\mathbf{w}_{ij}, \tag{4}$$

where $d_{ij}$ is the distance between the projector and the point. Then,

$$\left(\mathbf{x}'_P + d_{ij}\mathbf{w}_{ij}\right)^2 = 1, \tag{5}$$

so $d_{ij}$ is given by a second degree equation, whose physically relevant solution corresponds to the intersection point closer to the projector (assuming projection from outside the sphere):

$$d_{ij} = \mathbf{x}'_P \cdot \mathbf{w}_{ij} - \sqrt{(\mathbf{x}'_P \cdot \mathbf{w}_{ij})^2 - (\mathbf{x}'_P)^2 - 1}. \tag{6}$$

Performing the mapping requires solving Eq (6) for all the points of the projected image.

There are 6 unknown parameters: the three components of the position of the projector $\mathbf{x}_P$, the translation of the sphere relative to the center of the image, $\phi_0$ and $\theta_0$, and the projection scaling $\Delta\phi$.

**Gathering dataset for the mapping.** The positions of the projected pixels are measured with the camera using a structured light approach involving scanlines. The process is as follows:

1. A mask is generated to identify where the projection hits the bowl. A circular mask on the image by the projection of a fully white image;
2. The background subtractor is initialized with the projection of a black image. Multiple images are acquired and are then used to set up a background subtractor (using the MOG2 algorithm);
3. Images are captured with a projection of one line of one pixel every 10 pixels, horizontally, then vertically;

4. The background subtractor then the mask is applied to the images
5. For each combination of a horizontal line (corresponding to $i_P$) and a vertical line (corresponding to $j_P$), the two corresponding images are multiplied. The result highlights the single point of intersection; $(i_P, j_P)$;
6. Their positions on the image are then identified by blob detection;
7. Using the synthesized image of the bowl, the real position of each pixel, $\mathbf{x}_{ij}$, is recovered.

This dataset of correspondences between projected pixels $(i_P, j_P)$ and positions in real space $\mathbf{x}_{ij}$ is used to fit the mapping.

**Parameter Fitting.** To facilitate the convergence of the non-linear optimization routine used to fit the 6 projector parameters, reasonable initial estimates are generated using the collected data. We proceed as follows:

1. An initial guess for the projector's 3D position is made, typically based on the experimental setup, assuming, e.g., that the projector points towards the sphere's center from 1.5 m along the $z$-axis:

$$\mathbf{x}_P = (0\,\text{m}, 0\,\text{m}, 1.5\,\text{m}). \tag{7}$$

2. The centroid of the observed projected points is calculated, yielding the average location in projector pixel coordinates, $\mathbf{c} = (c_h, c_v)$, and its corresponding average 3D position, $\mathbf{x}_c$.
3. Potential small corrections $(\delta\phi, \delta\theta)$ can be added to account for $\mathbf{c}$ not being the exact optical center:

$$\phi_0 = c_h + \delta\theta, \quad \theta_0 = c_v + \delta\phi. \tag{8}$$

4. The angular size of a pixel, $\Delta\phi$ (assuming square pixels, $\Delta\theta = \Delta\phi$), is estimated using the spatial distribution of points around the central point $\mathbf{x}_c$. The average spatial distance $ds$ between adjacent measured points near $\mathbf{x}_c$ is calculated, and $\Delta\phi$ is approximated as the angle subtended by $ds$ at the estimated distance from the projector.

These initial estimates provide a starting point for the iterative optimization algorithm, increasing the likelihood of finding an accurate solution for the projector parameters.

**Identification of the water surface.** With the 3D depth camera, the position of the water surface can be easily identified. Once the bowl is filled with 15 L of water, a cardboard plate is put on the water's surface. Using the mask and a relevant threshold, the position of the cardboard plate can be easily retrieved on the 3D image. The associated point cloud is then fitted with a plane whose parameters are the distance from the water surface to the camera, and the direction of the normal of the plane (co-linear with gravity).

**Mapping.** From there, we can create a texture that maps each pixel of the projected image with the normal direction on the bowl. With this map, we need to render for each pixel the direction seen by the fish, taking into account its real spatial position. The process is as follows:

1. An omnidirectional camera render is obtained from the position of the fish. 6 cameras pointing in each direction of a cube are combined on a cubemap;
2. Using the texture map of the normal of the bowl, the direction seen by the individual is computed;
3. Those directions are used to render from the cubemap;
4. Finally, the rendering is then done on a texture, that is recorded by a full-HD orthographic camera.

There are additional constraints due to how the fish positions, handled by the CPU, are passed to the shader on the GPU. Data transfer from the CPU to the GPU is slow due to limited memory bandwidth. However, the shader does have

access to the position of the object it is attached to. Hence, the rendered texture is carried by the object moving according to the position of the fish. To get this whole process working correctly, the camera rendering to the cube map needs to remain static while all the objects are compensating for the movement of the scene.

The dimensioned plan as well as the list of items necessary for the construction of the main frame of the virtual reality are provided in Supplementary information (see S1 Text, S1 Table and S1 Fig). All software components required for the VR-system—including the acquisition and 3D tracking software, the trajectory simulator that controls virtual fish behavior, and the rendering software that displays virtual fish based on both simulated and real positions—are also provided as open-source tools.

This approach is intended to enable the widest possible adoption of this experimental technique to study animal behavior and analyze social interactions, fostering reusability, adaptability, and collaborative improvement within the scientific community.

## 2.5 Experimental procedure and data collection

### 2.5.1 Ethics statement.
Experiments were approved by the Animal Experimentation Ethics Committee C2EA-01 of the Toulouse Biology Research Federation and were performed in an approved fish facility (A3155501) under permit APAFIS#27303-2020090219529069 v8 in agreement with the French legislation. All procedures were designed to minimize stress and handling. Fish were transferred from rearing tanks to the virtual reality setup with minimal manipulation. Each individual was used in only one one-hour experimental session per week. Swimming ability was monitored throughout; fish exhibiting impaired or absent swimming activity were excluded and replaced. No animals were sacrificed during this study.

### 2.5.2 Study species.
Rummy-nose tetras (*Hemigrammus rhodostomus*) were purchased from Amazonie Labège in Toulouse, France. Fish were kept in 16 L aquariums on a 12:12 hour, dark:light photoperiod, at 24.9°C ($\pm$0.8°C) and were fed *ad libitum* with fish flakes. The average body length of the fish used in these experiments is 31 mm.

### 2.5.3 Experimental procedure.
The experiments reported in this article aim to validate the virtual reality device by demonstrating the effective nature of the interactions between the real fish and virtual fish and by quantifying the behavior of the real fish in response to three behavioral parameters of the virtual fish: speed, swimming depth, and swimming distance to the edge of the experimental bowl.

We first studied the impact on the fish behavior of a non-social control stimulus consisting of a black sphere. The sphere had a diameter of 3 cm, ensuring consistency with the dimensions of the original fish model, and was rendered as an opaque, dark-grey object. In this series of experiments, the same fish was first exposed for 30 minutes to a virtual fish moving along a uniform circular trajectory with constant depth (5 cm), speed (10 cm/s), and distance to the tank wall (10.4 cm). After 30 minutes, the realistic three-dimensional fish model was replaced with a spherical model following the same uniform circular trajectory. The experiment then continued for a further 30 minutes.

Seven conditions were then studied experimentally in which the virtual fish moves along a uniform circular trajectory with constant depth, speed, and distance to the edge of the tank (see Table 1). We also studied two additional conditions in which the virtual fish performs a rhodonea curve at a constant speed and constant depth.

We performed 7 one-hour repetitions with the non-social control stimulus, 10 one-hour repetitions for each condition in which the virtual fish performed circular trajectories, and 5 one-hour repetitions for each condition in which the virtual fish followed a rhodonea trajectory.

At the start of each trial, one fish was randomly removed from one of the breeding aquariums and placed in the hemispherical tank of the VR setup. The fish was then acclimatized to the new environment and lighting conditions during a period of 10 minutes before the trial started. The anamorphic image of the virtual fish is then displayed on the wall of the translucent dome in which the fish swims. The 3D tracking of the fish makes it possible to obtain its 3D position in the

**Table 1**. Speed, swimming depth, and distance from the wall of the virtual fish in each experimental condition.

| Experimental condition | Circular trajectories | | | | | | | rhodonea | | Control |
|---|---|---|---|---|---|---|---|---|---|---|
| | C1 | C2 | C3 | C6 | C7 | C8 | C9 | Rose 1 | Rose 2 | Virtual fish/sphere |
| Speed (cm/s) | 10 | 5 | 15 | 10 | 10 | 10 | 10 | 10 | 10 | 10 |
| Depth (cm) | 5 | 5 | 5 | 5 | 5 | 3.5 | 8 | 5 | 5 | 5 |
| Distance to the wall (cm) | 5.4 | 5.4 | 5.4 | 10.4 | 2.4 | 5.5 | 5 | 1.4 | 1.4 | 10.4 |
| Number of loops per rotation ($n$) | – | – | – | – | – | – | – | 3 | 3 | – |
| Total angular period ($p$) | – | – | – | – | – | – | – | 5 | 1 | – |

hemispherical tank, which feeds the model controlling the projected image in real time and which is also recorded at a frequency of 30 Hz. We then let the fish swim and interact with the virtual fish whose swimming speed, depth, and distance to the wall of the tank are kept at constant values, and only its anamorphic 3D projection changes in real time to take into account the relative position of the real fish compared to that of the virtual fish. Each fish tested under experimental conditions is used at most once a week and in a single trial to prevent any habituation phenomenon.

## 2.6 Data pre-processing

Before the data obtained during experiments can be analyzed, it must be preprocessed. This step ensures that only the periods of time during which fish are active are retained while eliminating errors from tracking. Thus, time intervals during which the fish remain motionless for at least 4 seconds are excluded from the analysis, as well as intervals when the fish's speed exceeds 25 cm/s, or when the fish swims in a straight line at a constant speed for at least one second.

Next, the data are filtered to remove any instances where the fish's position appears outside the bowl. To obtain trajectories with consecutive temporal data, gaps created by data removal are filled using linear interpolation. However, only gaps shorter than one second are filled this way. Beyond one second, this procedure can no longer be used to reconstruct the trajectory, which remains discontinuous at that point.

The trajectories obtained after this step thus consist of multiple segments within which the data remains consecutive. Each segment of a trajectory is then smoothed using a Gaussian kernel smoothing method, which helps reduce noise while preserving the essential features of the data,

$$x(n\Delta t) = \frac{\sum_{i=-n}^{n} x[(n+i)\Delta t] \exp\left(-\left(\frac{i\Delta t}{h}\right)^2\right)}{\sum_{i=-n}^{n} \exp\left(-\left(\frac{i\Delta t}{h}\right)^2\right)},$$
(9)

where $n = 4h/\Delta t$, so that the exponential is close to zero when $i = n$. The value of the parameter $h = 0.5$ s is chosen so that the smoothing is as low as possible in order not to average the data, while reducing noise as much as possible.

## 2.7 Data analyses

**2.7.1 Statistical metric for probability distribution comparison.** To quantitatively compare probability distribution functions (PDFs) obtained under different experimental conditions, we used the Hellinger distance as a measure of similarity [37,38]. Given two normalized PDFs, $F(x)$ and $G(x)$, corresponding to the same observable, the square of the Hellinger distance is defined as

$$H^2(F|G) = \frac{1}{2} \int \left(\sqrt{F(x)} - \sqrt{G(x)}\right)^2 dx = 1 - \int \sqrt{F(x)G(x)}\, dx.$$
(10)

This metric ranges from 0 to 1, where $H(F|G) = 0$ if and only if the two distributions are identical, and values approaching 1 indicate distributions with almost non-overlapping support. The second form of the equation provides a straightforward interpretation: it measures the deviation from unity of the scalar product between the square-rooted distributions, viewed as unit vectors in Euclidean space. In practice, a small Hellinger distance ($H < 0.2$) indicates a high similarity between the distributions, while values greater than 0.2 indicate a meaningful dissimilarity. This approach provides a robust and interpretable quantification of differences between behavioral distributions and has been successfully applied to assess the agreement between empirical fish data and model simulations in collective behavior studies.

**2.7.2 Correlation analyses.** In order to quantify how similar the dynamical behaviors of a real fish and a virtual object are, we calculate the cross-correlation function of their velocity vectors $\vec{v}_R$ and $\vec{v}_V$ with the following expression:

$$C_{RV}(\tau) = \frac{\langle \vec{v}_R(t) \cdot \vec{v}_V(t + \tau) \rangle}{\sqrt{\langle \vec{v}_R(t) \cdot \vec{v}_R(t) \rangle \langle \vec{v}_V(t) \cdot \vec{v}_V(t) \rangle}}. \tag{11}$$

Note that the velocity of reference is that of the real fish. As the virtual object often has a programmed periodic behavior (circular motion along circles of fixed radius and depth, or rhodonea of fixed parameters ($n,d$) and depth), this allows us to obtain a periodic correlation function and simplify the interpretation.

For circular trajectories, the period is simply given by $T = P/v$, where $P = 2\pi R$ is the perimeter, $R$ is the radius of the circle, and $v$ is the speed of the virtual object. For rhodonea trajectories, however, the perimeter cannot be expressed so directly; further details are provided in the Supplementary Information (S1 Text).

# 3 Results

This section presents a detailed analysis of the fish's response to the virtual fish along a variety of experimental conditions, with the aim of assessing the robustness and ecological validity of the setup.

## 3.1 Control condition with a non-social stimulus

We test a non-social control condition in which fish were exposed to a black sphere of similar size to conspecifics with a constant circular motion of radius $R = 10$ cm, speed $v = 10$ cm/s, and depth $z = 5$ cm (see S1 Video). Fig 6 shows that, unlike with virtual fish, individuals did not maintain close spatial proximity to the sphere nor exhibit alignment matching, confirming that the following responses observed in our VR assays are specific to social cues rather than generic reactions to moving objects.

Fig 6C shows the cross-correlation of the velocity vector of the real fish, $\vec{v}_R(t)$, with that of the virtual object, $\vec{v}_V(t + \tau)$, for both cases: the virtual fish and the virtual sphere, calculated using Eq 11. The reference speed is that of the real fish. Since the virtual object repeats the same state (position and velocity) with a period $T = 2\pi R/v$, both correlation functions are periodic with this same period. When the virtual object represents a conspecific, the maximum correlation is nearly three times higher ($C_{fish}^{max} \approx 0.77$) than when it represents a non-social object ($C_{sphere}^{max} \approx 0.27$), demonstrating unequivocally that the fish is far more responsive to the image of a conspecific.

The maxima are reached with a slightly negative delay, $\tau_{fish} = -0.133$ s $+nT$ and $\tau_{sphere} = -0.067$ s $+nT$, with $T = 6.28$ s and $n = 0, \pm 1, \pm 2, \dots$. This indicates that the real fish tends to follow the virtual conspecific with a short delay of about 0.13 s, whereas with the virtual sphere the delay is negligible but the correlation remains very low, confirming the reduced attraction to the non-social stimulus.

## 3.2 Behavioral response of real fish to the circular movement of a virtual fish

Fig 7 shows some of the different cases analyzed in detail in this section. Each panel represents the trajectories of both the real and the virtual fish, in red and blue, respectively, over one minute. Time is represented on the vertical axis and

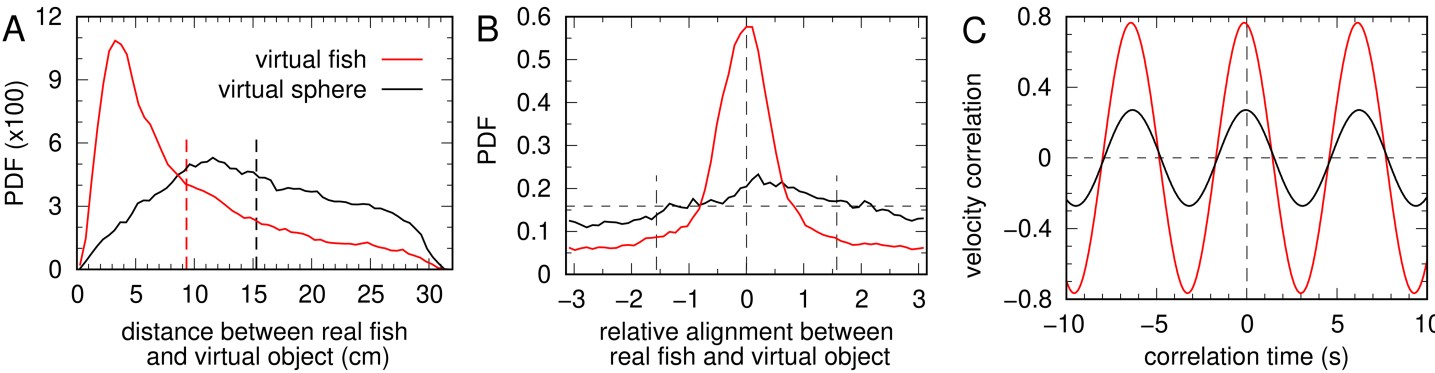

**Fig 6**. **Comparison of fish responses to a virtual conspecific versus a virtual sphere.** (A) Probability density function (PDF) of the distance between the real fish and the virtual fish (red, mean 9.3 cm ± 7.0 SD) and between the real fish and a virtual sphere (black, mean 15.3 cm ± 7.2 SD). Vertical lines indicate the mean value of each distribution. (B) PDF of the relative alignment of the real fish with the virtual fish (red) and with the virtual sphere (black). (C) The maximum correlation with the virtual fish is $C \approx 0.77$, reached at $\tau = -0.133\,\mathrm{s} + nT$, and $C \approx 0.27$ with the virtual sphere, reached at $\tau = -0.067 + nT$, with $T = 6.28\,\mathrm{s}$, $n = 0, \pm1, \pm2, \dots$

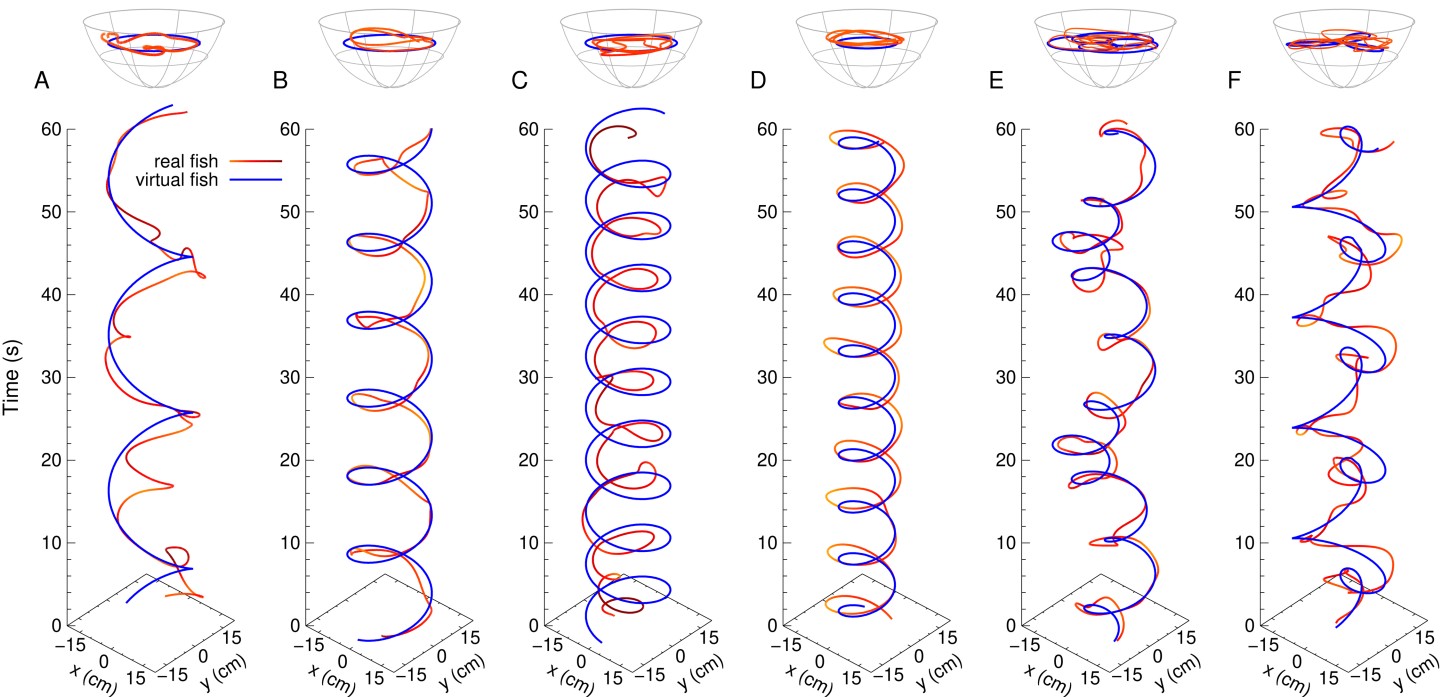

**Fig 7**. **Real and virtual fish trajectories.** Trajectories of the real fish (red) and the virtual fish (blue) over one minute when the virtual fish swims at a constant speed at depth $z = 5$ cm. (A–C) the virtual fish follows horizontal circular paths at a fixed distance from the wall, $r_w = 5.4$ cm: (A) v = 5 cm/s, (B) v = 10 cm/s, and (C) v = 15 cm/s. (D) Circular paths of smaller radius: v = 10 cm/s, $r_w = 10.4$ cm. (E) Rose curves given by $r = \rho \cos(n\theta/d)$, with $\rho = 19$ cm, $n = 3$, $d = 5$, and (F) with $n = 3$, $d = 1$.

grows upwards, horizontal position is shown in *xy*-coordinates, and color indicates depth (which varies little across cases). In all cases, the trajectories clearly show that the real fish reacts to the virtual one as if it were a real conspecific, staying close and accompanying its motion throughout the bowl.

While the first three panels (Fig 7ABC) show that the real fish is able to appropriately follow the projected image on the nearest part of the bowl, the fourth panel (Fig 7D) reveals a crucial aspect of the setup: the real fish positions itself between the virtual fish and the edge of the bowl (see also Fig 8 and S2 Video). This means that the fish is responding to the anamorphic image projected on the opposite side of the screen, moreover, through the water, as if the virtual conspecific was perceived as being clearly inside the bowl —a remarkable phenomenon, given the significant distortion introduced by the water.

This observation is significant for two reasons. First, it confirms that the optical setup generates a convincing illusion of the presence of a conspecific. Second, it shows that the fish interprets the projected image in a biologically meaningful way: it responds not merely to local visual cues, but to the coherent representation of both the typical size and behavior of a conspecific, maintaining typical distances and relative positioning as it would with a real partner.

We now take a closer look at each condition to see how the fish's response depends on the different parameters. We impose a constant circular motion on the virtual fish and focus on three parameters characterizing its behavior: the speed $v_{VF}$, the depth $z_{VF}$, and the horizontal distance to the edge of the bowl, denoted by $r_w^{VF}$. The latter refers to the *distance to the wall*, commonly used in studies of fish behavior in quasi two-dimensional setups. As we will show, real fish tend to preserve their swimming depth, making it more appropriate to characterize the proximity to the bowl's *wall* by using the horizontal distance rather than the cylindrical distance from the edge of the sphere.

We perform three sets of experiments, varying one parameter at a time along three values, while keeping the others constant. Note that the intermediate case is the same in the three cases.

**3.2.1 Impact of the virtual fish's swimming speed on the behavior of the real fish.** We first examine the effect of varying the swimming speed. Three swimming speeds of the virtual fish are considered: $v_{VF} = 5$, 10, and 15 cm/s, while the depth and the distance to the wall are kept constant at $z_{VF} = 5$ cm and $r_w^{VF} = 5.4$ cm respectively. These cases correspond to the trajectories shown in Figs 7ABC.

In each case, the real fish adjusts well to the virtual fish, with the peak of the probability density function (PDF) of its speed coinciding with the value of the virtual speed for the two slower conditions (Fig 9B and S2 Table). The mean speeds of the real fish are 6.3, 9.4, and 11.7 cm/s, respectively. For the slowest case, the real fish tends to be faster than the VF,

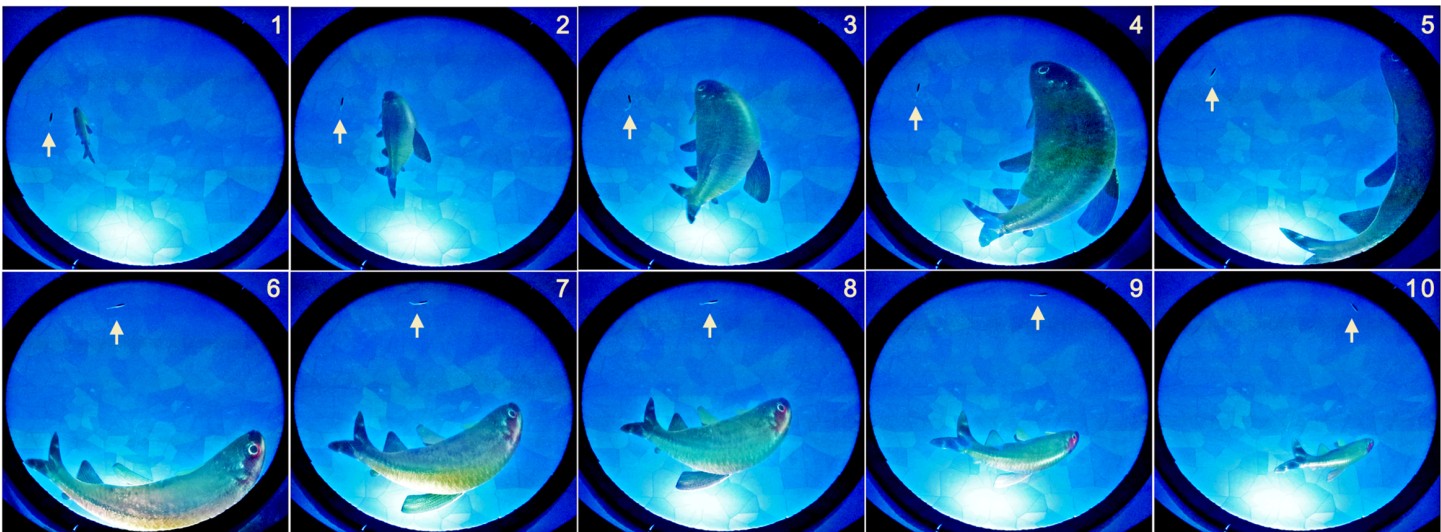

**Fig 8**. **A 3-second sequence showing the real fish reacting to the anamorphic projection of the virtual conspecific.** The appearance and size of the virtual fish are adapted in real time so that the real fish (indicated by a white arrow pointing to its position) swimming with the edge of the bowl on its left has the illusion that the virtual fish is swimming nearby and to its right.

while for the highest speed, the PDF peak shifts slightly to the left, around 14 cm/s, and a secondary bump appears near 5 cm/s, indicating occasional much slower responses, which could be due to a loss of contact or to a difficulty to maintain such a high speed.

The distance of the real fish to the virtual one remains consistent across the three cases, with the peak of the distance PDF located at $d \approx 2.5$ cm (Fig 9A and S3 Table), although the long tail in the three cases shows that fish can be quite distant. Similarly, the real fish maintains a nearly constant depth of approximately $z \approx 4.5$ cm, slightly above the virtual fish (Fig 9C and S3 Table), but occasionally falls at $z_{RF} \approx 10$ cm.

Figs 9D and 9E show the relative position of the real fish in the reference frame of the virtual fish, placed at the origin and with its velocity vector pointing towards the positive values of the $y$-axis. Figs 9D corresponds to the $xy$-plane, while Figs 9E shows the $xz$-plane. These plots confirm that the real fish stays close to the virtual fish: at low speed, it is slightly in front; at medium speed, it is equally likely to be ahead or behind; and at high speed, it is often behind and slightly closer to the center of the bowl, i.e., farther from the wall. In all cases, the real fish tends to swim slightly above the virtual fish. At high speed, some trajectories extend as deep as 6 cm below the virtual fish.

S2 Fig shows the cross-correlation of the velocity vector of the real fish, $\vec{v}_R(t)$, with that of the virtual fish, $\vec{v}_V(t + \tau)$, for the three tested speeds. For $v_{VF} = 10$ cm/s and $v_{VF} = 15$ cm/s, the correlations are high, comparable to those observed in Fig 6C when the fish interacts with the virtual conspecific. Even for $v_{VF} = 5$ cm/s, the correlation remains well above the values obtained with a virtual sphere. The maximum correlations are $C_1^{max} \approx 0.71$, $C_2^{max} \approx 0.54$, and $C_3^{max} \approx 0.79$, reached respectively at $\tau_1 = 0$ s $+nT_1$, $\tau_2 = 0.4$ s $+nT_2$, and $\tau_3 = -0.267$ s $+nT_3$, with $T_1 = 3\pi$ s, $T_2 = 6\pi$ s, and $T_3 = 2\pi$ s.

At $v_{VF} = 10$ cm/s, the real fish closely follows the virtual one, showing a strong correlation. The delay is negligible, which is consistent with the fact that this speed is close to the typical mean speed of rummy-nose tetra [10]. For $v_{VF} = 5$ cm/s, the virtual fish moves too slowly, and the real fish often overshoots it, as indicated by the positive delay of nearly half a second. For $v_{VF} = 15$ cm/s, the real fish tend to follow the virtual fish with a delay of more than a quarter of a second, still maintaining a high correlation that reflects consistent following behavior.

In light of these results, the trajectories shown in Fig 7ABC can be interpreted as follows: when the virtual fish swims at 5 cm/s, the real fish is faster and circles around it, moving back and forth as if waiting. When the virtual fish swims at 15 cm/s, the real fish generally follows but tends to shortcut, tracing smaller circles and occasionally skipping a full turn to wait for the next pass of the virtual fish. The intermediate speed of 10 cm/s appears to correspond to the comfort speed of the real fish when interacting with a virtual conspecific.

These results show that the real fish is strongly attracted by the virtual fish, and consistently seeks to stay nearby.

**3.2.2 Impact of virtual fish's swimming depth on the behavior of the real fish.** We now consider three depths of the virtual fish, $z_{VF} = 3.5$, 5, and 8 cm, and constant speed and distance to the wall $v_{VF} = 10$ cm/s and $r_w = 5.4$ cm respectively. In all three cases, the real fish maintains the same average distance to the virtual one, as shown in the three PDFs of the distance between fish, all sharply peaked at $d \approx 2.5$ cm (Fig 10A and S3 Table). Similarly, the real fish closely matches the speed of the virtual fish, with the three PDFs nearly symmetric and centered at $v_{VF}$ (Fig 10B and S3 Table). The same imitation pattern is observed in the PDFs of the depth (Fig 10C and S3 Table), with the real fish swimming practically at the same depth as the virtual one, slightly above it (as already observed in the speed variation experiment), except in the deepest case. There, the fish's maneuverability and range of depth choices is limited by the reduced vertical space, only 3.5 cm, while the body height of *H. rhodostomus* is approximately 0.5 cm. Despite that, the variation towards greater depths remains similar across all three cases, with variability in depth selection rarely exceeding $\pm 1.5$ cm.

The deeper the virtual fish swims, the less space the real fish has to stay at the same depth. Thus, the greater the depth, the smaller the average distance at which the real fish stays from the virtual one: 7.97, 7.83, and 6.25 cm, for $z_{VF} =$ 3.5, 5, and 8 cm, respectively. The $x$-range around the virtual fish covered by the real fish, indicated by the radius of the density maps in Fig 10EF, reaches around 38 cm in the less deep cases, but only 15 cm when the virtual fish is at 8 cm depth (see the region of high density in blue and red in the color maps). Despite this, the peak of the PDF of the distance

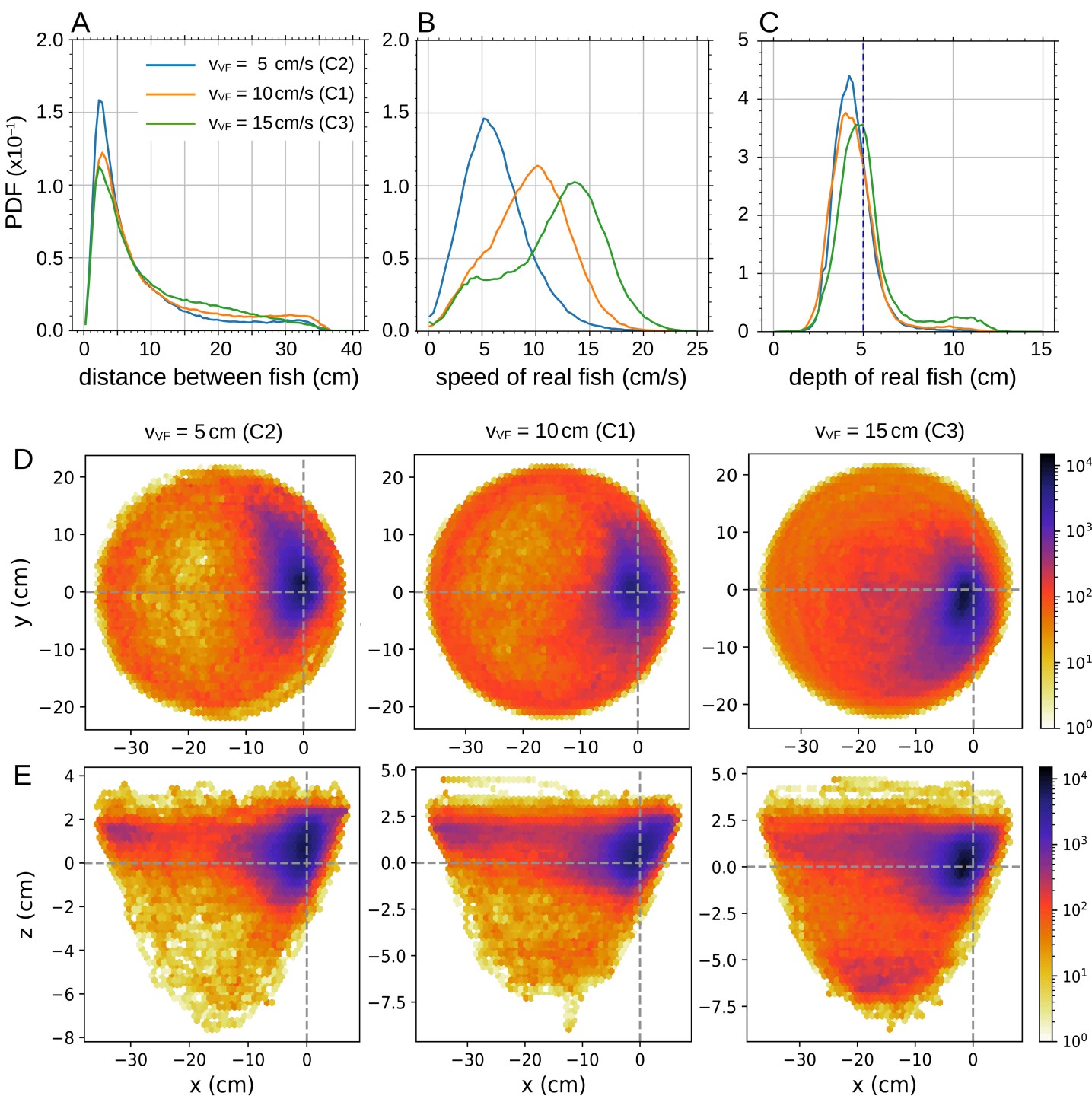

**Fig 9**. **Impact of virtual fish's swimming speed on real fish's behavior.** Probability density functions (PDF) of (A) the distance between the real fish and the virtual fish, (B) the swimming speed of the real fish, and (C) the swimming depth of the real fish, for 3 values of the **swimming speed** of the virtual fish: $v_{VF} = 5$, 10, and 15 cm/s. The virtual fish (VF) swims at constant depth and distance to the wall in all cases, $z_{VF} = 5$ cm and $r_w^{VF} = 5.4$ cm respectively. (DE) Relative position of the real fish in the reference frame of the virtual fish for the three swimming depths, in (D) the $xy$-plane and (E) the $xz$-plane. The virtual fish is located at (0,0). The $y$-axis points in the direction of the fish, and the $x$-axis is perpendicular to the fish and points outward from the bowl.

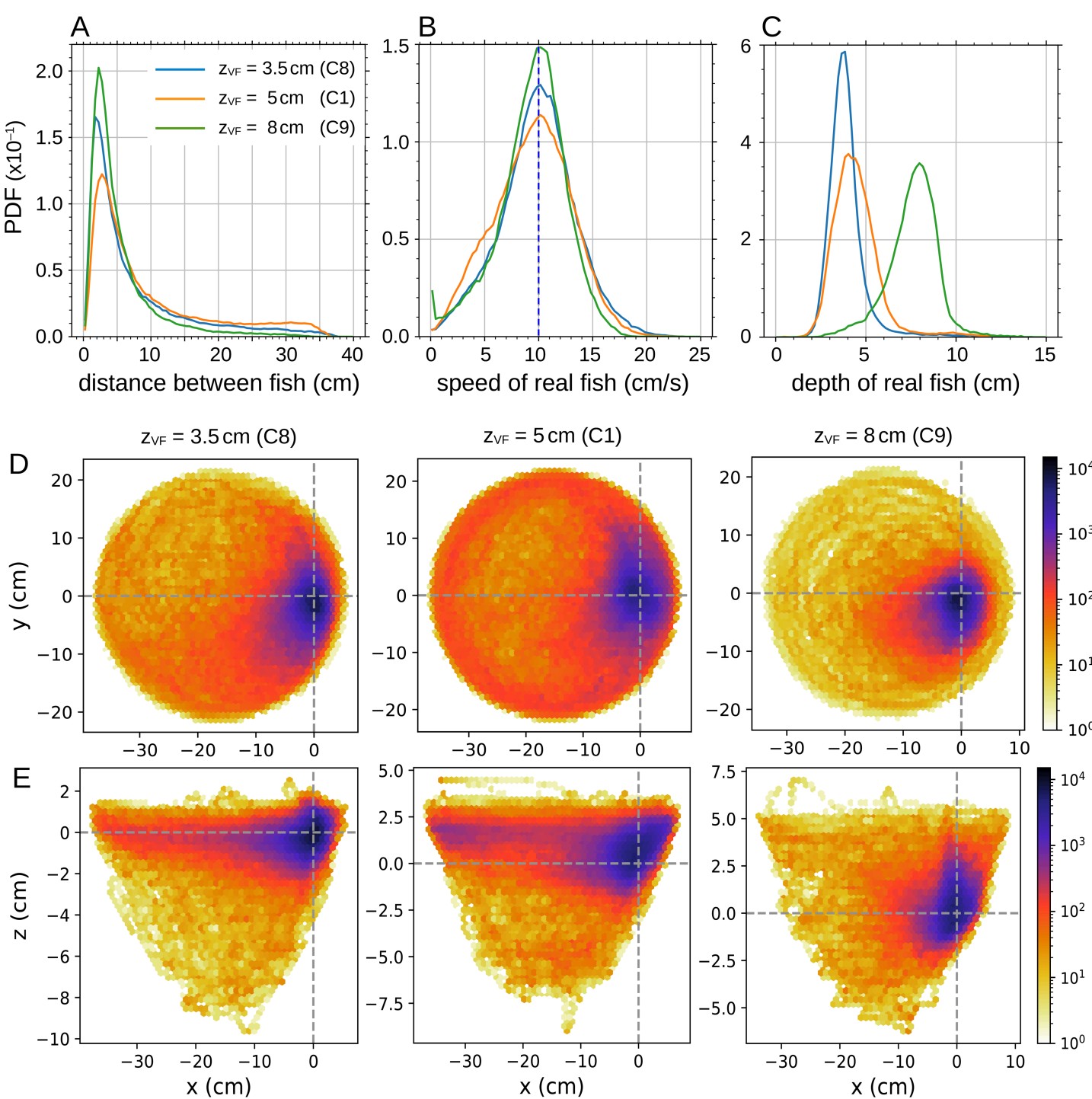

**Fig 10**. **Impact of virtual fish's swimming depth on real fish's behavior.** Probability density functions (PDF) of (A) the distance between the real fish and the virtual fish, (B) the swimming speed of the real fish, and (C) the swimming depth of the real fish, for 3 values of the **swimming depth** of the virtual fish: $z_{VF}$ = 3.5, 5, and 8 cm/s. The virtual fish (VF) swims at a constant speed $v_{VF}$ = 10 cm/s in all cases, and at a distance from the wall $r_w^{VF}$ = 5.5, 5.4, and 5 cm for each depth, respectively. (DE) Relative position of the real fish in the reference frame of the virtual fish for the three swimming depths, in (D) the *xy*-plane and (E) the *xz*-plane. The virtual fish is located at (0,0). The *y*-axis points in the direction of the fish, and the *x*-axis is perpendicular to the fish and points outward from the bowl.

between fish remains unchanged across depths, indicating that the real fish maintains the same typical distance to the virtual fish. Finally, the asymmetry of the PDF of the depth $z_{VF} = 8$ cm, with a longer tail toward deeper values, is explained by the limited space available below the virtual fish at that depth.

**3.2.3 Impact of the virtual fish's distance to the wall on the behavior of the real fish.** In the third experiment, we used three values of the distance between the virtual fish and the wall, $r_w^{VF} = 2.4$, 5.4, and 10.4 cm, while keeping constant the virtual fish's speed and depth at $v_{VF} = 10$ cm/s and $z_{VF} = 5$ cm respectively.

As in the previous cases, the real fish tends to adopt the same behavioral patterns as the virtual fish when the setup geometry allows it. The separation between fish is again the same in the three conditions, with the peak of the three PDFs of the distance between fish consistently located at 2.5 cm (Fig 11A and S4 Table). The velocity and depth of the real fish are again symmetric curves respectively centered in $v_{VF}$ (Fig 11B and S4 Table) and $z_{VF}$ (Fig 11C and S4 Table), with, as before, a slight preference for being positioned above the virtual fish.

The high peak of the PDF of the distance between fish when the virtual fish is farther from the wall is due to the fact that in that case the maximum available distance for the real fish is reduced. The diameter of the small circle of the sphere at depth $z = 5$ cm is approximately 40 cm; since the virtual fish is at about 10 cm from the wall, the real fish can only move as far as about 30 cm. This is also visible in the radius of occupancy of the real fish around the virtual one (Fig 11E). When the virtual fish is the closest to the wall, there is practically no room for the real fish between the virtual one and the edge of the bowl. In contrast, when the virtual fish swims at 10.4 cm from the border, the real fish stays more frequently closer to the virtual one (the color map far from the virtual fish is clearer than when $r_w^{VF} = 5$ cm), and often swims in the region between the virtual fish and the wall, as shown in Fig 7D, demonstrating the effectiveness of the anamorphic projection on the opposite side of the bowl.

Similarly to what was observed in the condition closest to the surface, the fact that the virtual fish is very close to the limits of the swimming space (delimited by the edge of the bowl and the surface of the water) does not perturb the preference of the real fish when it is positioned at the side of the virtual fish where these limits have no effect.

## 3.3 Behavioral response of real fish to the motion of a virtual fish along rose-like trajectories

We now explore how the real fish responds to more complex trajectories of the virtual fish, specifically, two rhodonea trajectories given by the parametric equation $r = \rho \cos(n\theta/d)$, where $(r, \theta)$ are the polar coordinates, with $\rho = 19$ cm, $n = 3$ and $d = 5$ (Fig 7E) and $n = 3$ and $d = 1$ (Fig 7F and S3 Video). We fix the speed and depth of the virtual fish to $v_V = 10$ cm/s and $z_V = 5$ cm respectively, which are the intermediate values used in the previous experiments in circles, and the shortest distance to the wall to $r_w^V = 1.4$ cm (so the maximum radius is $R_V = \rho = 19$ cm). The interest lies in how the fish reacts as the virtual fish transitions from moving along the edge of the bowl to traveling through its interior, where water distortion can have stronger effects on the image perceived by the fish, and with a varying curvature of the trajectory, particularly in the $d = 1$ case, where the petals are sharp and pointed, introducing abrupt changes on the projected image of the virtual fish. These trajectories are particularly demanding for the fish, as they often cross the central part of the bowl, where the anamorphic image projected on the screen is subject to considerable optical distortion due to the wider amount of water from the edge of the bowl.

Despite this challenge, the density map of Rose 1 (Fig 12A) clearly matches the trajectory of the virtual fish, with three broad and smooth lobes corresponding to the three rose petals. This indicates that the fish is able to accurately follow the virtual fish even under strong visual distortion. In contrast, the trajectory described in Rose 2 (Fig 12D) has narrower petals, requiring sharper turns from the real fish, and the density map is less defined, suggesting that the real fish has more difficulty in following the virtual one.

In Rose 2, the green areas indicate that the fish spends more time near the tips of the petals, where it slows down to perform the turning maneuvers (Fig 12D). The central green region is simply explained by the geometry of the trajectory:

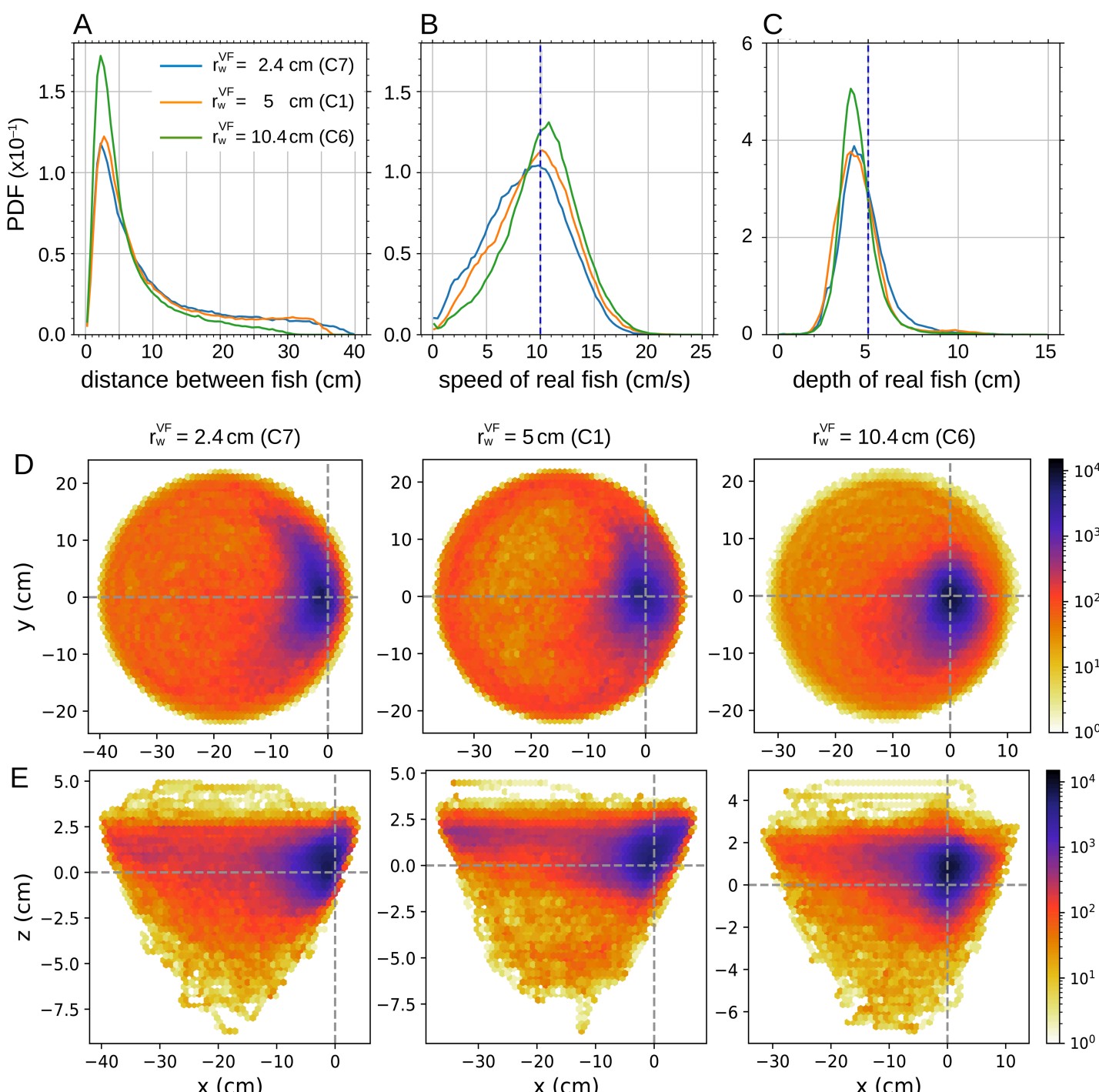

**Fig 11**. **Impact of virtual fish's distance to the wall on real fish's behavior.** Probability density functions (PDF) of (A) the distance between the real fish and the virtual fish, (B) the swimming speed of the real fish, and (C) the swimming depth of the real fish, for 3 values of the **distance to the wall** of the virtual fish: $r_w^{VF} = 2.4$, 5.4, and 10.4 cm. The virtual fish (VF) swims at constant speed and depth in all cases, $v_{VF} = 10$ cm/s and $z_{VF} = 5$ cm respectively. (DE) Relative position of the real fish in the reference frame of the virtual fish for the three swimming depths, in (D) the $xy$-plane and (E) the $xz$-plane. The virtual fish is located at (0,0). The $y$-axis points in the direction of the fish, and the $x$-axis is perpendicular to the fish and points outward from the bowl.

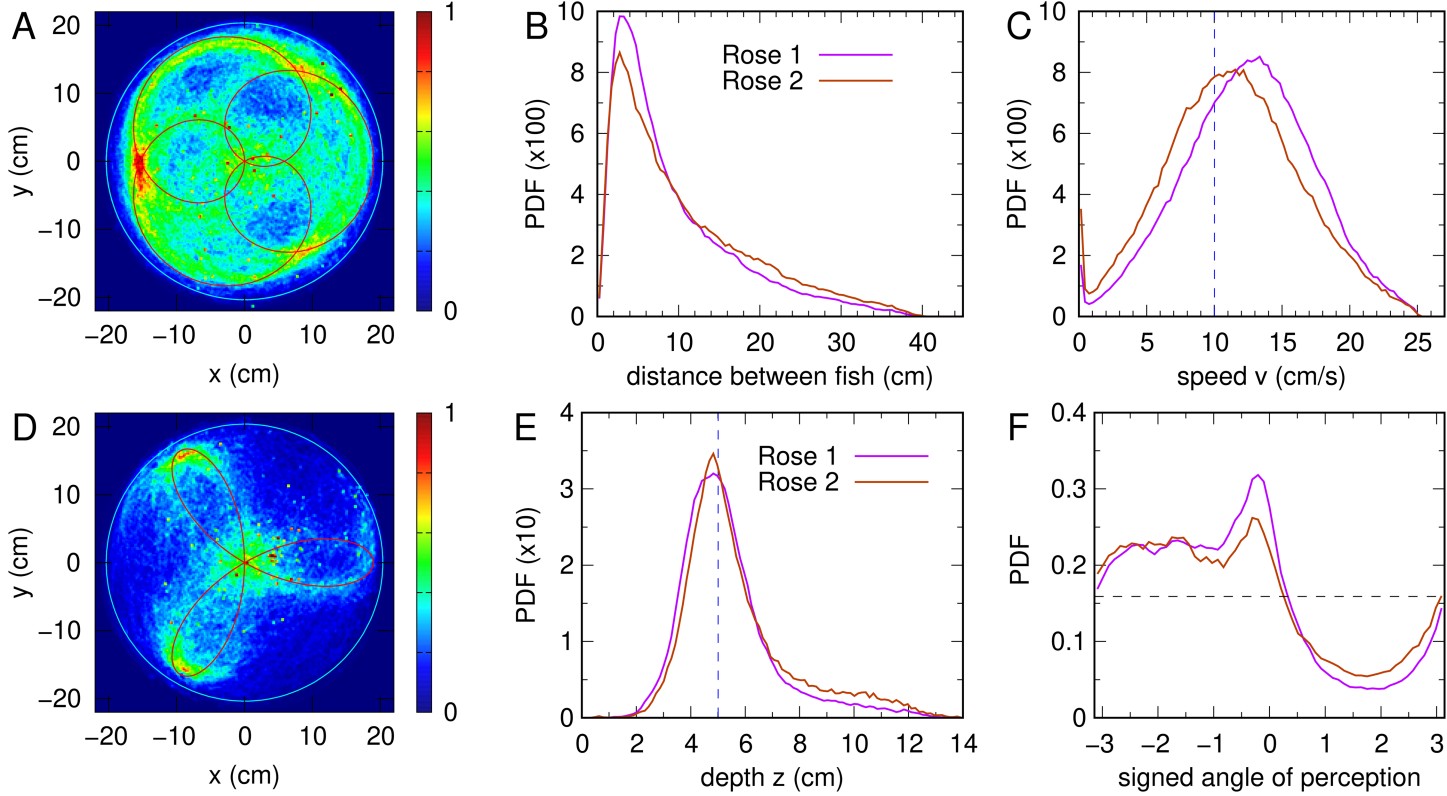

**Fig 12.** **Behavior of real fish when following the virtual one along rhodonea trajectories.** (AD) Rhodonea trajectories of three petals. Red line: trajectory of the virtual fish. Density map: positions of the real fish. (A) Rose 1, wide petals ($n = 3$, $d = 5$), (D) Rose 2, narrow petals ($n = 3$, $d = 1$). (B) Probability density function (PDF) of the distance between fish along Rose 1 (magenta) and Rose 2 (brown) trajectories. (C) PDF of the speed of the real fish in each rose. Vertical blue dashed line: speed of the virtual fish. (E) PDF of the depth of the real fish in each rose. Vertical blue dashed line: depth of the virtual fish. (F) PDF of the angle with which the virtual fish is perceived by the real one, multiplied by the sign of the virtual fish direction of rotation (−1 for clockwise, +1 for counterclockwise).

the fish crosses the center three times more often than the outer tips, a pattern that also happens in the red regions of the density map of Rose 1.

In both roses, the real fish maintains similar distances to the virtual fish as those observed in circular trajectories, with a peak of the PDF at the same value, 2.5 cm, although of smaller size (Fig 12B). This means that the real fish swims slightly farther away than in the circular cases. Comparing the roses, there is a slight tendency to keep a greater distance in Rose 2, likely due to the sharper turns.

Also, in both roses, the fish is able to adjust to the speed of the virtual fish, with both PDFs of the speed centered on $v_{VF}$, but wider than in the circular case, showing that following these irregular paths is more demanding in terms of speed adjustment (Fig 12C). Moreover, the peak of the curve of Rose 1 is further to the right, indicating that the fish swims faster when following smoother turns.

Finally, there are no significant differences between the PDFs of the depth for the two roses; both curves are almost symmetric and centered very slightly to the left of the expected depth of 5 cm (Fig 12E).

Fig 12F shows the PDF of the angle $\psi^+$, defined as the angle with which the virtual fish is perceived by the real one, multiplied by the sign of the direction of rotation of the virtual fish (−1 clockwise, +1 counterclockwise). In both roses, once the direction of rotation is given, the virtual fish always has the wall at the same side along the whole path of the rose: to its left during clockwise motion, and to its right during anticlockwise motion. This allows us to cumulate the experiments

performed in both directions and reveal the lateral positioning of the virtual fish with respect to the real one. Both PDFs show that the real fish tends to position itself so that the virtual fish remains in front of it, as shown by the peaks at $\psi^+ \approx 0$. More interestingly, both PDFs are quite asymmetric, revealing a robust behavioral preference: the real fish tends to position itself on the inner side of the path of the virtual fish. In other words, regardless of the rotation direction, the real fish places itself on the side of the virtual fish towards which the latter is turning.

S3 Fig shows the cross-correlation $C_{RV}(\tau)$ between the velocity vectors of the real fish, $\vec{v}_R(t)$, and the virtual fish, $\vec{v}_V(t + \tau)$, in both rhodonea conditions. As already observed, despite the complexity of the trajectories, the correlation values remain high, only slightly lower than those observed for circular trajectories. The maximum correlation is higher for Rose 1 ($C_{max} \approx 0.65$) than for Rose 2 ($C_{max} \approx 0.51$), confirming that Rose 2 is more difficult for the real fish to follow. The respective maxima are reached with a slight negative delay at $t_1 = -0.23$ s ($+nT_1$, $T_1 = 24.6$ s) and $t_2 = -0.1$ s ($+nT_2$, $T_2 = 13.4$ s), for $n = 0, \pm 1, \pm 2, ...$, indicating that in both cases, the real fish tends to follow the virtual one. In Rose 1, this delay is longer (almost a quarter of a second) but yields a stronger correlation, whereas in Rose 2, the delay is shorter (one tenth of a second), but the correlation is weaker.

## 4 Discussion

Understanding how animals interact socially is a long-standing goal in behavioral research [1,2]. However, studying these processes in real time is often difficult. Traditional methods struggle to capture the complexity of live interactions and cannot easily manipulate environmental factors. In this work, we present and validate an open-source and low-cost immersive virtual reality (VR) system designed specifically to investigate real-time social interactions between fish. We show that this system allows a real fish to swim freely while interacting with a computer-generated virtual fish whose behavior can be precisely controlled. The VR setup uses closed-loop feedback, meaning that the virtual fish responds to the real fish's movements in real time.

Our results show that real fish adjust their swimming speed depending on the speed of the virtual fish. When the virtual fish swims slowly, the real fish often moves faster, sometimes circling and waiting for it. At intermediate speeds, the real fish synchronizes more closely, and at higher speeds, it sometimes struggles to keep up, resulting in shortcuts or pauses. This shows that real fish are sensitive to the movement speed of their virtual partners and modify their own behavior accordingly. It also suggests that real fish prefer to maintain proximity to conspecifics moving at naturalistic speeds. Previous studies in live groups also indicated that matching speed is critical for coordination, and these results strengthen that conclusion by isolating the interaction to a single partner [39,40].

We also find that real fish tended to swim at about the same depth as the virtual fish, usually slightly above it. Even when the virtual fish swims deeper or closer to the surface, the real fish adapts remarkably well. However, when the virtual fish moves very close to the bottom, real fish have less vertical room to maneuver, resulting in slightly reduced matching. These results demonstrate the real fish's strong ability to track vertical position, even under spatial constraints. To determine their swimming depth, it has been shown that fish use the rate in change of pressure mediated by changes in swim-bladder volume, together with their vertical speed [41].

Our results further reveal that the spatial context, particularly the distance of the virtual fish to the tank wall, significantly shapes real fish behavior. When the virtual fish swims closer to the wall, real fish follows it tightly. When the virtual fish swims farther from the wall, the real fish maintains the same close distance but sometimes maneuvers between the virtual fish and the wall. This shows that real fish perceive and adapt to environmental constraints while trying to stay close to a social partner. Moreover, these results highlight the effectiveness of the anamorphic projection: fish behaved as if the virtual conspecific was physically present in the water, even when distortions could have been expected. This spatial flexibility echoes findings from studies on collective navigation in open water, where environmental features modulate but do not eliminate social attraction [42].

Finally, our results demonstrate that real fish were able to follow complicated paths such as rhodonea (non-circular) trajectories surprisingly well. When petals were broad and smooth, fish tracked the virtual fish closely. When the path

required sharper turns, fish performance declined somewhat, with real fish lagging or showing wider turns. Still, the basic tendency to follow remained strong. This indicates that the VR setup can be used to study not only simple pursuit but also more intricate forms of social tracking. These results suggest that fish can integrate more complex motion cues than previously thought. In natural settings, this ability likely supports following in rapidly changing environments, such as evading predators or tracking group members in turbulent flows.

## 5 Conclusions

In summary, our study shows that a simple, low-cost, and open-source VR system can effectively create realistic social interactions between real and virtual fish. Real fish responded dynamically to changes in virtual fish speed, depth, distance from the wall, and trajectory complexity. They showed remarkable behavioral flexibility and consistency, even under visual distortions or challenging movement patterns. The system's closed-loop design was essential, ensuring that virtual stimuli adapted to real fish movements, making the interactions genuinely interactive.

The current work focuses on open-loop validation, but the design also supports closed-loop integration with three-dimensional behavioral models, enabling causal tests of social interaction rules. Moreover, the ability to project multiple virtual fish offers new opportunities to study how individuals integrate information from several neighbors and how local rules scale to group dynamics [28,43,44].

This platform opens new possibilities for studying how sensory inputs drive social decisions at fine scales. It may also serve as a model for future work combining VR, brain imaging, and genetic tools to uncover the neural basis of collective behavior [45,46]. It demonstrates how low-cost, accessible technologies can significantly advance our understanding of animal social cognition. Ultimately, our findings may have broader implications for the fields of robotics, artificial intelligence, and swarm intelligence, where biological principles of collective motion inspire novel engineering solutions [47,48].

## Supporting information

**S1 Text. Experimental setup, software architecture, and trajectory design for the closed-loop virtual reality system.**
(PDF)

**S1 Fig. Experimental setup.** Descriptive diagram (left) and dimensions (right).
(PDF)

**S2 Fig. Cross-correlation of the velocity vector of the real fish with that of the virtual fish at different virtual fish speeds.** Virtual fish conditions correspond to C1, C2, and C3, with $v_1 = 10$ (orange), $v_2 = 5$ (blue), and $v_3 = 15$ cm/s (green). The trajectories of the virtual fish are circles of radius $R = 15$ cm at depth $z = 5$ cm corresponding to a distance to the wall of $r_w = 5.4$ cm. Maximum correlation is reached at $t_1 = 0$ s $+nT_1$, $t_2 = 0.4$ s $+nT_2$, and $t_3 = -0.267$ s $+nT_3$, where $T_1 = 3\pi$ s, $T_2 = 6\pi$ s, and $T_3 = 2\pi$ s, for $n = 0, \pm1, \pm2, \dots$.
(PDF)

**S3 Fig. Cross-correlation of the velocity vectors of the real and virtual fish for the two rhodonea trajectories.** In both cases, the swimming speed is $v = 10$ cm/s, the minimum distance to the wall is $r_w = 1.4$ cm, and the swimming depth is $z = 5$ cm (corresponding to a maximum radius of $R = 19$ cm). The maximum correlation for Rose 1 (purple) is reached at $t_1 = -0.23$ s $+nT_1$, with $T_1 = 24.6$ s, and for Rose 2 (brown) at $t_2 = -0.1$ s $+nT_2$, with $T_2 = 13.4$ s, for $n = 0, \pm1, \pm2, \dots$. Vertical dashed lines indicate the respective period lengths $T_1$ and $T_2$.
(PDF)

**S1 Table. Main frame materials of the Virtual Reality setup.**
(PDF)

**S2 Table. Hellinger distance between the probability distribution function (PDF) of the 3 observables used to quantify the impact of virtual fish's swimming speed on real fish's behavior.** We list the distances obtained for the three conditions compared pairwise. The real fish maintained the same swimming distance and depth relative to the virtual fish across the three swimming speeds tested (i.e., no significant differences were found between the PDFs). However, the swimming speeds of the real fish were significantly different between the three conditions C1, C2, and C3 (i.e., when the virtual fish moved at 5, 10, and 15 cm/s respectively). Values of Hellinger distance are shown in bold font when $H > 0.2$ (high dissimilarity of the PDFs).
(PDF)

**S3 Table. Hellinger distance between the probability distribution function (PDF) of the 3 observables used to quantify the impact of virtual fish's swimming depth on real fish's behavior.** We list the distances obtained for the three conditions compared pairwise. The swimming distance of the real fish to the virtual fish was significantly smaller when the virtual fish was at 8 cm depth. There was also a significant difference in the swimming depth of the real fish between the three conditions C1, C8, and C9. Values of Hellinger distance are shown in bold font when $H > 0.2$ (high dissimilarity of the PDFs).
(PDF)

**S4 Table. Hellinger distance between the probability distribution function (PDF) of the 3 observables used to quantify the impact of the virtual fish's distance to the wall on the real fish's behavior.** We list the distances obtained for the three conditions compared pairwise. The swimming speed and depth of the real fish were not influenced by the virtual fish's proximity to the tank wall. Only when the virtual fish was swimming close to the wall at 2.4 cm in condition C7, the real fish maintained a higher distance in comparison with condition C6, in which the distance to the wall of the virtual fish was much larger (10.4 cm). Values of Hellinger distance are shown in bold font when $H > 0.2$ (high dissimilarity of the PDFs).
(PDF)

**S1 Video. Comparison of a fish's behavior in the presence of a virtual fish and then a virtual ball under control conditions.** Video excerpts of an experiment with a real fish interacting with the anamorphic projection of a virtual conspecific during 30 minutes and then with a non-social control stimulus consisting of a black sphere during the next 30 minutes in the experimental bowl of radius 250 mm. Both the virtual fish and virtual sphere move along a uniform circular trajectory with a constant speed of 10 m/s, at a constant depth of 5 cm below the water surface, and at a constant distance of 10.4 cm to the edge of the tank. The video shows, after one minute, the replacement of the three-dimensional fish model with the spherical model. Top-Left: user interface allowing the real-time visualization of the trajectories of the real and virtual fish (in the *xy*- and *xz*-planes) and the instantaneous modification of the parameters of the model driving the virtual fish. Bottom-Left: real-time 3D tracking of the real fish. Right panel: anamorphic rendering of the virtual fish projected onto the acrylic bowl by the rendering application according to the 3D position of the real fish.
(MP4)

**S2 Video. Behavioral response of a real fish to the circular movement of a virtual fish.** Video excerpts of an experiment with a real fish interacting with the anamorphic projection of a virtual conspecific in the experimental bowl of radius 250 mm. The virtual fish moves along a uniform circular trajectory with a constant speed of 10 cm/s, at a constant depth of 5 cm below the water surface, and at a constant distance of 5.4 cm to the edge of the tank. Top-Left: user interface allowing the real-time visualization of the trajectories of the real and virtual fish (in the *xy*- and *xz*-planes) and the instantaneous modification of the parameters of the model driving the virtual fish. Bottom-Left: real-time 3D tracking of the real fish. Right panel: anamorphic rendering of the virtual fish projected onto the acrylic bowl by the rendering application according to the 3D position of the real fish.
(MP4)

**S3 Video. Behavioral response of real fish to the motion of a virtual fish along a rose-like trajectory.** Video excerpt of an experiment with a real fish interacting with the anamorphic projection of a virtual conspecific in the experimental bowl of radius 250 mm. The virtual fish moves along a rhodonea (rose-like) trajectory with a constant speed of 10 cm/s and at a constant depth of 5 cm below the water surface. The rhodonea trajectory is given by the parametric equation $r = \rho \cos(n\theta/d)$, where $(r, \theta)$ are the polar coordinates, with $\rho = 19$ cm, $n = 3$ and $d = 1$. Top-Left: user interface allowing the real time visualization of the trajectories of the real and virtual fish (in the $xy$- and $xz$-planes) and the instantaneous modification of the parameters of the model driving the virtual fish. Bottom-Left: real-time 3D tracking of the real fish. Right panel: anamorphic rendering of the virtual fish projected onto the acrylic bowl by the rendering application according to the 3D position of the real fish.
(MP4)

## Acknowledgments

The authors warmly thank Zhen Kang for his help in processing the control condition data.

## Author contributions

**Conceptualization:** Stéphane Sanchez, Clément Sire, Guy Theraulaz.

**Data curation:** Ramón Escobedo.

**Formal analysis:** Ramón Escobedo, Audrey Denis, Clément Sire, Guy Theraulaz.

**Funding acquisition:** Stéphane Sanchez, Clément Sire, Guy Theraulaz.

**Investigation:** Stéphane Sanchez, Ramón Escobedo, Audrey Denis, Guy Theraulaz.

**Methodology:** Stéphane Sanchez, Ramón Escobedo, Renaud Bastien, Clément Sire, Guy Theraulaz.

**Project administration:** Guy Theraulaz.

**Resources:** Renaud Bastien, Mathieu Moreau, Maud Combe.

**Software:** Stéphane Sanchez, Ramón Escobedo, Renaud Bastien, Boris Lenseigne, Mathieu Moreau, Maud Combe, Andrew D. Straw.

**Supervision:** Stéphane Sanchez, Clément Sire, Guy Theraulaz.

**Validation:** Ramón Escobedo, Boris Lenseigne, Clément Sire, Guy Theraulaz.

**Visualization:** Stéphane Sanchez, Renaud Bastien, Mathieu Moreau, Maud Combe.

**Writing – original draft:** Stéphane Sanchez, Ramón Escobedo, Renaud Bastien, Guy Theraulaz.

**Writing – review & editing:** Stéphane Sanchez, Ramón Escobedo, Clément Sire, Guy Theraulaz.

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
