## [Decision Letter · Decision Letter 0]

10 Sep 2025

PONE-D-25-36359A low-cost closed-loop Virtual Reality system to investigate social interactions and collective behavior in fishPLOS ONE

Dear Dr. Theraulaz,

Thank you for submitting your manuscript to PLOS ONE. After careful consideration, we feel that it has merit but does not fully meet PLOS ONE’s publication criteria as it currently stands. Therefore, we invite you to submit a revised version of the manuscript that addresses the points raised during the review process.

We look forward to receiving your revised manuscript.

Kind regards,

Shicheng Li

Academic Editor

PLOS ONE

**Journal Requirements:**

1. When submitting your revision, we need you to address these additional requirements. Please ensure that your manuscript meets PLOS ONE's style requirements, including those for file naming. The PLOS ONE style templates can be found at https://journals.plos.org/plosone/s/file?id=wjVg/PLOSOne_formatting_sample_main_body.pdf and https://journals.plos.org/plosone/s/file?id=ba62/PLOSOne_formatting_sample_title_authors_affiliations.pdf 2. Please update your submission to use the PLOS LaTeX template. The template and more information on our requirements for LaTeX submissions can be found at http://journals.plos.org/plosone/s/latex. 3. Please include a complete copy of PLOS’ questionnaire on inclusivity in global research in your revised manuscript. Our policy for research in this area aims to improve transparency in the reporting of research performed outside of researchers’ own country or community. The policy applies to researchers who have travelled to a different country to conduct research, research with Indigenous populations or their lands, and research on cultural artefacts. The questionnaire can also be requested at the journal’s discretion for any other submissions, even if these conditions are not met.  Please find more information on the policy and a link to download a blank copy of the questionnaire here: https://journals.plos.org/plosone/s/best-practices-in-research-reporting. Please upload a completed version of your questionnaire as Supporting Information when you resubmit your manuscript. 4. To comply with PLOS One submissions requirements, in your Methods section, please provide additional information regarding the experiments involving animals and ensure you have included details on (a) methods of sacrifice, (b) methods of anesthesia and/or analgesia, and (c) efforts to alleviate suffering. 5. Thank you for stating the following financial disclosure: Agence Nationale de la Recherche (ANR-20-CE45-0006-1)Spanish AEI grant PID2020-115088RB-I00   Please state what role the funders took in the study.  If the funders had no role, please state: "The funders had no role in study design, data collection and analysis, decision to publish, or preparation of the manuscript." If this statement is not correct you must amend it as needed. Please include this amended Role of Funder statement in your cover letter; we will change the online submission form on your behalf. 6. Thank you for uploading your study's underlying data set. Unfortunately, the repository you have noted in your Data Availability statement does not qualify as an acceptable data repository according to PLOS's standards. At this time, please upload the minimal data set necessary to replicate your study's findings to a stable, public repository (such as figshare or Dryad) and provide us with the relevant URLs, DOIs, or accession numbers that may be used to access these data. For a list of recommended repositories and additional information on PLOS standards for data deposition, please see https://journals.plos.org/plosone/s/recommended-repositories. 7. Your ethics statement should only appear in the Methods section of your manuscript. If your ethics statement is written in any section besides the Methods, please move it to the Methods section and delete it from any other section. Please ensure that your ethics statement is included in your manuscript, as the ethics statement entered into the online submission form will not be published alongside your manuscript. 8. Please include captions for your Supporting Information files at the end of your manuscript, and update any in-text citations to match accordingly. Please see our Supporting Information guidelines for more information: http://journals.plos.org/plosone/s/supporting-information. 9. If the reviewer comments include a recommendation to cite specific previously published works, please review and evaluate these publications to determine whether they are relevant and should be cited. There is no requirement to cite these works unless the editor has indicated otherwise. 

Reviewers' comments:

Reviewer's Responses to Questions

**Comments to the Author**

1. Is the manuscript technically sound, and do the data support the conclusions?

Reviewer #1: Yes

Reviewer #2: Yes

2. Has the statistical analysis been performed appropriately and rigorously?

Reviewer #1: Yes

Reviewer #2: Yes

3. Have the authors made all data underlying the findings in their manuscript fully available?

Reviewer #1: Yes

Reviewer #2: Yes

4. Is the manuscript presented in an intelligible fashion and written in standard English?

Reviewer #1: Yes

Reviewer #2: Yes

5. Review Comments to the Author

**Reviewer #1:** This article sounds very interesting and impressive. Written in good English language.

However, the only flaw that requires major revision and amendment is the related works regarding this study.

The Result subheading shown the analysis of the experiment followed by discussion.

The methodology section needs to be reorganized prior the result and analysis section.

Conclusion section is not found. Is it replaced with discussion?

**Reviewer #2:** This paper presents a major methodological contribution by developing a low-cost, open-source, closed-loop virtual reality system that allows the study of social interactions in fish with unprecedented control. The results convincingly demonstrate that real fish perceive and follow virtual fish as credible conspecifics, adjusting their speed, depth, and relative position in a flexible and robust manner. The strength of the study lies in its experimental rigor, the quality of the quantitative data, and the full transparency of the shared codes and data. However, the statistical analysis would benefit from including inferential tests to formally support comparisons between conditions. Furthermore, while the system allows for visual closed-loop, the behavior of the virtual fish remains pre-programmed in an open loop in this study, thus limiting the exploration of more complex bidirectional interactions. Finally, although the method is powerful for deciphering individual rules, it does not yet address group dynamics involving multiple real individuals.

To significantly strengthen the statistical robustness of the evaluation, several additional analyses are possible. The use of non-parametric Kruskal-Wallis tests, followed by Dunn post-hoc tests, would allow for a formal comparison of the distributions of speed, distance, and depth between the different experimental conditions. In addition, calculating cross-correlations between the speeds and directions of the real and virtual fish would objectively quantify their degree of synchronization and response times. Finally, the addition of a control condition with a non-social stimulus would ensure that the observed behaviors are indeed specific social responses and not simple reactions to a movement.

6. PLOS authors have the option to publish the peer review history of their article (what does this mean?). If published, this will include your full peer review and any attached files.

Reviewer #1: No

Reviewer #2: No

---

## [Author Response · Author response to Decision Letter 1]

10 Oct 2025

Responses to Reviewers

Reviewers have raised a total of 6 points that we reproduce here in black italicized text. Our answers are given in blue.

Response to the comments of Referee 1

This article sounds very interesting and impressive. Written in good English language.

#1.1: However, the only flaw that requires major revision and amendment is the related works regarding this study.

A: We appreciate the Reviewer’s positive assessment and constructive requests. In the revised version of our paper, we have incorporated additional references (19 to 25) to create a comprehensive and thematically organized Related Work section that more clearly positions our contribution within the existing literature.

#1.2: The Result subheading shown the analysis of the experiment followed by discussion. The methodology section needs to be reorganized prior the result and analysis section.

A: We have reordered the Methods section before the Results section thus following the PloS One template.

#1.3: Conclusion section is not found. Is it replaced with discussion?

A: We added a concise conclusion. We hope these revisions make the manuscript clearer, better contextualized, and easier to navigate.

Response to the comments of Referee 2

This paper presents a major methodological contribution by developing a low-cost, open-source, closed-loop virtual reality system that allows the study of social interactions in fish with unprecedented control. The results convincingly demonstrate that real fish perceive and follow virtual fish as credible conspecifics, adjusting their speed, depth, and relative position in a flexible and robust manner. The strength of the study lies in its experimental rigor, the quality of the quantitative data, and the full transparency of the shared codes and data. However, the statistical analysis would benefit from including inferential tests to formally support comparisons between conditions. Furthermore, while the system allows for visual closed-loop, the behavior of the virtual fish remains pre-programmed in an open loop in this study, thus limiting the exploration of more complex bidirectional interactions. Finally, although the method is powerful for deciphering individual rules, it does not yet address group dynamics involving multiple real individuals.

A: We sincerely thank Reviewer 2 for his/her positive and encouraging assessment of our work. We are pleased that the manuscript was recognized as presenting a major methodological contribution, with rigorous experiments, transparent data sharing, and convincing demonstrations of the ability of real fish to treat virtual fish as credible conspecifics. In the revised manuscript, we have also added a specific measure of the dissimilarity between the PDFs to formally compare behavioral responses across conditions, and the corresponding results are now reported in the Results section. Below we address the two other constructive points raised, and explain how we have revised the manuscript accordingly. Open-loop behavior of the virtual fish. We agree that in the initial submission the virtual fish followed pre-programmed trajectories. Our revised manuscript clarifies that this study aimed to validate the platform in open-loop conditions, while emphasizing that the system is designed for full closed-loop experiments. We now highlight ongoing work in which the trajectory of the virtual fish is controlled in real time by three-dimensional interaction models derived from the computational analysis of real fish pairs. These future experiments will allow bidirectional interactions and causal tests of social rules, which we will report in a separate article. Extension to group dynamics. We acknowledge that the present study also focused on interactions between one real and one virtual fish. The revised version now explicitly outlines how the system will be scaled to project multiple virtual conspecifics that respond dynamically to each other and to the real fish. This will allow tests of how individuals combine information from multiple neighbors and how local rules scale to group-level coordination, directly connecting to predictions from recent modeling work (Lei et al., 2020; Wang et al., 2022; Xue et al., 2023}.

#2.1: To significantly strengthen the statistical robustness of the evaluation, several additional analyses are possible. The use of non-parametric Kruskal-Wallis tests, followed by Dunn post-hoc tests, would allow for a formal comparison of the distributions of speed, distance, and depth between the different experimental conditions.

A: To address this point, we have now incorporated an additional subsection entitled “Statistical Metric for Probability Distribution Comparison” in the Data Analyses section. In this new analysis, we quantify differences between behavioral distributions observed under distinct experimental conditions using the Hellinger distance, a widely used and robust statistical metric for comparing probability distribution functions. Specifically, given two normalized distributions, the Hellinger distance provides a bounded measure of similarity ranging from 0 (identical distributions) to 1 (non-overlapping distributions). Small values indicate strong similarity, whereas values greater than 0.2 reveal marked differences. This approach allows us to rigorously assess whether variations in fish responses across experimental manipulations (e.g., changes in speed, depth, or distance to the wall of the virtual fish) correspond to statistically meaningful shifts in the underlying behavioral distributions. We also clarify in the revised manuscript how these analyses complement the descriptive statistics already presented, thereby reinforcing the robustness of our findings. By adding this inferential framework, we ensure that our conclusions are not only visually convincing but also formally supported by statistical tests.

#2.2: In addition, calculating cross-correlations between the speeds and directions of the real and virtual fish would objectively quantify their degree of synchronization and response times.

A: As suggested, we have calculated the cross-temporal correlations between the velocities of the real and virtual objects (a fish or a black sphere) to provide an additional quantification of synchronization and response times. The results fully confirm our previous conclusions. They can be found, together with the figures, in the following parts of the text:

- In Sec. 3.8.2, we use the formula (11) to calculate the cross-correlation, where the real fish is taken as the reference of C(\tau).

- In Sec. 4.1 and Fig. 6C, showing that the correlation with a virtual conspecific is almost three times larger than with a virtual sphere, showing a clear preference for the social stimulus.

- In Sec. 4.2.1 and S2 Fig., we show that in the three cases with different speeds, the values of the correlation remain high, with the real fish anticipating the virtual one at low speed, behaving closely at intermediate values of the speed, and following the virtual fish with a short delay at high speed.

- In Sec. 4.3 and S3 Fig., for the case of the two rhodonea, correlations are again high. The real fish follows the virtual one more consistently in Rose 1 than along Rose 2, as the correlation is higher and the delay longer and negative in Rose 1, where the correlation is higher and the delay longer (about 0.23 s, negative), whereas in Rose 2 the delay is almost zero but the correlation is weaker.

#2.3: Finally, the addition of a control condition with a non-social stimulus would ensure that the observed behaviors are indeed specific social responses and not simple reactions to a movement.

A: We thank the Reviewer for her/his suggestion. We have now included a non-social control stimulus consisting of a black sphere of comparable size to the tested fish. The new results show that fish did not follow or adjust their trajectories relative to the sphere, in sharp contrast to their robust following of virtual conspecifics. This confirms that the observed behaviors are indeed specific to social stimuli and not mere responses to moving objects. The revised manuscript includes these data in the Results section 4.1 (Control condition with a non-social stimulus) and discusses their implications in confirming the social specificity of the responses to 3D avatars.

Cited References :

Xue, T., Li, X., Lin, G., Escobedo, R., Han, Z., Chen, X., Sire, C. & Theraulaz, G. 2023. Tuning social interactions' strength drives collective response to light intensity in schooling fish. Plos Computational Biology, 19(11):e1011636.

Wang, W, Escobedo, R., Sanchez, S., Sire, C., Han, Z. & Theraulaz, G. 2022. The impact of individual perceptual and cognitive factors on collective states in a data-driven fish school model. Plos Computational Biology, 18: e1009437.

Lei, L., Escobedo, R., Sire, C., Theraulaz, G. 2020. Computational and robotic modeling reveal parsimonious combinations of interactions between individuals in schooling fish. Plos Computational Biology, 16: e1007194.

---

## [Editor Report · Decision Letter 1]

14 Dec 2025

An open-source closed-loop Virtual Reality system to investigate social interactions and collective behavior in fish

PONE-D-25-36359R1

Dear Dr. Theraulaz,

We’re pleased to inform you that your manuscript has been judged scientifically suitable for publication and will be formally accepted for publication once it meets all outstanding technical requirements.

Kind regards,

Shicheng Li

Academic Editor

PLOS One
---

## [Editor Report · Acceptance letter]

PONE-D-25-36359R1

PLOS One

Dear Dr. Theraulaz,

I'm pleased to inform you that your manuscript has been deemed suitable for publication in PLOS One. Congratulations! Your manuscript is now being handed over to our production team.

Kind regards,

on behalf of

Dr. Shicheng Li

Academic Editor

PLOS One